# Association of overtime work hours with various stress responses in 59,021 Japanese workers: Retrospective cross-sectional study

**Hiroyuki Kikuchi**[1,2], **Yuko Odagiri**[1,2]*, **Yumiko Ohya**[1], **Yutaka Nakanishi**[2], **Teruichi Shimomitsu**[1,2,3], **Töres Theorell**[4], **Shigeru Inoue**[1,2]

**1** Department of Preventive Medicine and Public Health, Tokyo Medical University, Shinjuku-ku, Tokyo, Japan, **2** Health Promotion Center, Public Health Research Foundation, Chiyoda-ku, Tokyo, Japan, **3** Japan Health Promotion and Fitness Foundation, Minato-ku, Tokyo, Japan, **4** Stress Research Institute, Stockholm University, Stockholm, Sweden

* odagiri@tokyo-med.ac.jp

**Data Availability Statement:** The data set supporting these findings is not publicly available due to access restrictions imposed by the Tokyo Medical University Ethics Committee. Public data

## Abstract

This study aims to clarify the relationships between length of overtime work and various stress responses using large-scale cross-sectional data of Japanese workers. This study's participants are 59,021 Japanese workers in 117 companies. Data was collected by self-reporting questionnaire. The Brief Job Stress Questionnaire was used to measure stress responses on six scales (i.e. "lack of vigor", "irritability", "fatigue", "anxiety", "depression", and "somatic responses"). Length of overtime work hours were classified as 0–20, 21–30, 31–40, 41–50, 51–60, 61–70, 71–80, and >80 hours/month. Multiple linear regression analyses were used to examine the association of stress responses with overtime while adjusting all possible confounders. In result, workers with longer overtime showed significantly higher "irritability", "fatigue", "anxiety", "depression", and "somatic responses" for both genders (p-for-trend <0.001), however, length of overtime was negatively associated with "lack of vigor" among men (p-for-trend <0.001). Men with 61–80 hours of overtime showed high fatigue with high vigor at the same time. Length of overtime was linearly associated with various stress responses, except for "lack of vigor". Length of overtime shows linear associations with various psychosomatic stress responses. However, "lack of vigor" was not consistently associated with overtime. Male workers with 61–80 hours of monthly overtime were more likely to feel vigorous than workers with shorter overtime. However, potential longterm effects of such extreme overtime should not be underestimated and must be paid attention to.

## Introduction

Long working hours are shown to affect various health outcomes [1,2], especially in cardiovascular diseases [3–6]. In addition, exceedingly long working hours also deteriorate worker's mental health causing issues such as alcoholism [7], and sleep disturbances [8]. Avoiding long working is needed for maintaining workers health.

sharing is restricted in order to protect privacy and confidentiality. Data requests from any interested researcher may be sent to the corresponding author (YO): odagiri@tokyo-med.ac.jp or our department prev-med@tokyo-med.ac.jp. The data set name for the study is PHR2017.

**Funding:** The author(s) received no specific funding for this work.

**Competing interests:** The authors have declared that no competing interests exist.

Especially in Japan, long working hours have been regarded as a serious social and health issue. *Karoshi* (sudden death caused by cardiovascular or cerebrovascular disease due to overwork) and *Karojisatsu* (suicide due to overwork) have gathered public attention from the mid-1970s [9,10]. Efforts to prevent *Karoshi* and *Karojisatsu* by limiting working hours have been attempted, however, the number of claimed and compensated cases of occupational mental disorders are substantially increasing [11,12]. In 2017, the Japanese government issued "*The Action Plan for the Realization of Work Style Reform*", which introduced an overtime limit (i.e. less than 100 hours/month) by law [13], in addition to upper limit of regular working hours (40 hours/week). However, much controversy has risen as to whether 100 hours/month is appropriate or not.

In this context, it is important to consider the upper overtime-limit to prevent mental illness among workers. So far, several studies have investigated associations between long working hours and mental health [1,14]. However, many studies used tertile [15,16], quartile [17–19] or quintile [20,21] cut-off were used for evaluating effects of amount of overtime work. Studies in which a broader range of overtime is used are needed for the establishment of an upper overtime-limit as a health policy. In addition, there are only few published in which gender-stratified analysis has been performed. This is due to the limited number of women with long working hours [19,22]. Labor force participation among women of working age increased substantially worldwide [23]. Furthermore, occupational class might influence the relation between overtime and health outcome [24]. To nuance the picture of associations between overtime and mental health, stratified analysis by these factors is also needed.

This study aims to clarify the detailed relationships between length of overtime and different stress responses using large size data of Japanese workers.

## Materials and methods

### Study design

This study is a cross-sectional study using data which was acquired by the Stress Check Program of Japan.

### Japan's stress check program

Detailed procedures of the Stress Check Program are described elsewhere [25,26]. Briefly, the program was started by an amendment of the Industrial Safety and Health Law in 2015. The program aims at primary prevention of mental health disorders, by reminding the worker of his/her own stress condition and facilitating the improvement of workplace environments. The program requires all workplaces with 50 or more employees to conduct a questionnaire survey regarding psychosocial stress at work at least once a year. Feedback is provided to each employee with aims to decrease the risk of mental health problems through enhancing their awareness of their own stress [26]. The law requires employers to provide all regularly hired employees with opportunities to participate in the Stress Check Program. According to the protocol issued by the Japanese government, employers are encouraged to remind non-responders to complete the Stress Check Program at least one time [25].

### Participants and data collection

The eligible participants of this study were 95,004 employees in 223 companies which implemented the Stress Check Program from December 2015 to November 2016, based on Industrial Health and Safety Law. The program was provided to the employer by one health-service company through commission of the employers. The health service company has branches located in four major cities (Sapporo, Tokyo, Osaka and Fukuoka). Based on contracts with

each company, they provide the Stress Check Program services to 223 companies, which are located in every prefecture of Japan.

The program was carried out via internet or written questionnaire.

This research was based on data collected as part of the Stress Check Program retrospectively. The primary objective of main project is investigating longitudinal effects of long working hours on workers mental and physical health. Data from participants who declined use of their data for research was excluded. In addition, workers who belong to workplaces with less than 50 employees were excluded because such workplaces have different occupational healthcare requirements under Japanese law. For example, assignment of occupational health physicians is not mandatory for such small workplaces. Furthermore, it is also not mandatory for them to conduct Stress Check Program, therefore, we excluded their data to ensure representability. Shift workers and part-time workers were excluded because their work schedule varies considerably different from other workers.

All data were collected in a non-anonymized manner to give feedback to participants in Stress Check Program, however the data was anonymized before being provided to the authors due to ethical issue.

## Independent variables: Length of overtime work hours

This study assessed the length of participants' monthly length of overtime work hours. Self-reported monthly overtime data was collected in increments of every 10 hours from "20 hours or less" to "141 hours or more". Due to the smaller number of participants who engaged in 81 hours or more overtime we consolidated them into one category, and thus the present study set the categorization of overtime as the following 8 categories, "20 hours or less", "21–30 hours", "31–40 hours", "41–50 hours", "51–60 hours", "61–70 hours", "71–80 hours" and "81 hours or more".

## Dependent variables: Stress responses

Stress responses were assessed using the Brief Job Stress Questionnaire (BJSQ) following the recommended protocol of the Stress Check Program [25–27]. The BJSQ was originally designed to measure both psychological and somatic, both positive and negative stress responses among workers in any workplace with minimum number of items [28]. Six stress-response scales can be measured by 29 questionnaire items in the BJSQ, and each items were developed by referring some already standardized/authorized questionnaires. In detail, "vigor", "fatigue", and "irritability", consisting of 3 items each, were from the Profile Of Mood States (POMS). "Depression", consisting of 6 items, was from the Center for Epidemiologic Studies for Depression Scale (CES-D). "Anxiety", consisting of 3 items, was from the State-Trail Anxiety Inventory (STAI). "Somatic stress responses", consisting of 11 items, was from Screener for the Somatoform Disorders and the Subjective Wellbeing Inventory (SUBI) [28]. Each item in the BJSQ asks the respondent to choose one of four options on the Likert scale. For example, to assess their vigor level, participants were given the phrase "*I have been very active*", and then asked to choose one of four options (i.e., "almost never", "sometimes", "often" and "almost always"). The total score on each scale was calculated by summing all items according to established protocol [25]. Then, standardized scores were derived for each scale of stress response. With regards to vigor score, we calculated "lack of vigor" score in this study by subtracting vigor score from 15. Higher scores indicate higher stress responses.

## Covariates

We collected data of gender, age, type of job (sales/professional/clerk/services/manual/transport/other), job class (regular/manager/director), employment status (regular/contract/

temporary/other) and type of schedule (inflexible/flexible/other) by self-report. In addition, company industry (construction/manufacturing/electricity, gas, heat supply and water/information and communications/transport and postal activities/trade/finance/scientific and development research institutes/accommodations, eating and drinking services/education/medical service and social welfare/public sector/other) and size (51-100/101-300/301-999/1,000–2,999/3,000 or more employees) were also obtained through open information. Job control, supervisor's support and coworker's support were also collected from questions on the BJSQ.

## Statistical procedure

Multiple linear regression analysis was used to examine the associations of stress responses with overtime. In each model, we used a standardized score of each stress response (such as "lack of vigor" or "irritability", etc.) as a dependent variable, "length of overtime work hours" as an independent variable, other individual characteristics (such as "age" and "job position", etc.) as covariates. First, we performed linear trend tests by treating "length of overtime work hours" as a continuous variable to check the linear association between each stress response and overtime. Then, to seek possible thresholds of overtime, "length of overtime work hours" were treated as a dummy variable by setting "less than 20hours" group as a reference category.

For sensitivity analysis, a multiple imputations technique was used because approximately 10% (n = 8,560) of subjects included missing data for any one of the analyzed variables [29]. We created 20 multiple imputed data sets which included all measurement variables using a multivariate normal imputation method under a missing at random assumption, and combined the estimated parameters using Rubin's combination methods [30].

Finally, stratified analyses by gender, age-category (i.e. less than 30, 30–59 and 60 or older) and job-position (regular and manager) were also performed. In these models, interaction terms between age-category/job-position and overtime work hours were used to explore possible effect modification.

All statistical tests were two-tailed and considered to be statistically significant at the 0.05 level. All analyses were done with Stata, version 15.0.

## Ethics

We acquired the written consent for the secondary-use of their data from each participant. In detail, participants were asked "I admit utilizing of my data for academic research purposes." with "Yes/No" options. Furthermore, the protocol of this study was opened to them through the website, and we accepted their refusal for their participation at any time. This protocol is in line with the Japanese Ethical Guidelines for Epidemiological Research, which regulates that informed consent is not necessarily required for observational studies utilizing the existing data [31]. All procedures of this study received approval from the Tokyo Medical University Ethics Committee (No. 2016–166).

## Results and discussions

Fig 1 shows the flow of study participants. By Industrial Health and Safety Law, 95,004 regular workers were offered to receive stress check program at their workplace from December 2015 to November 2016. After excluding 6,016 participants who did not receive the program and 5,518 participants who declined secondary use of their data, the data-set of 83,470 participants was initially established (mean [± standard deviation] age; 44.0 [±11.2] years old, age range: 17–89 years old, 62.7% men). Then, we excluded participants at workplaces with less than 50 employees (n = 921), part-time workers (n = 5,039), shift workers (n = 9,929) and those with any missing data (n = 8,560) from the data-set. Some participants had two or more conditions

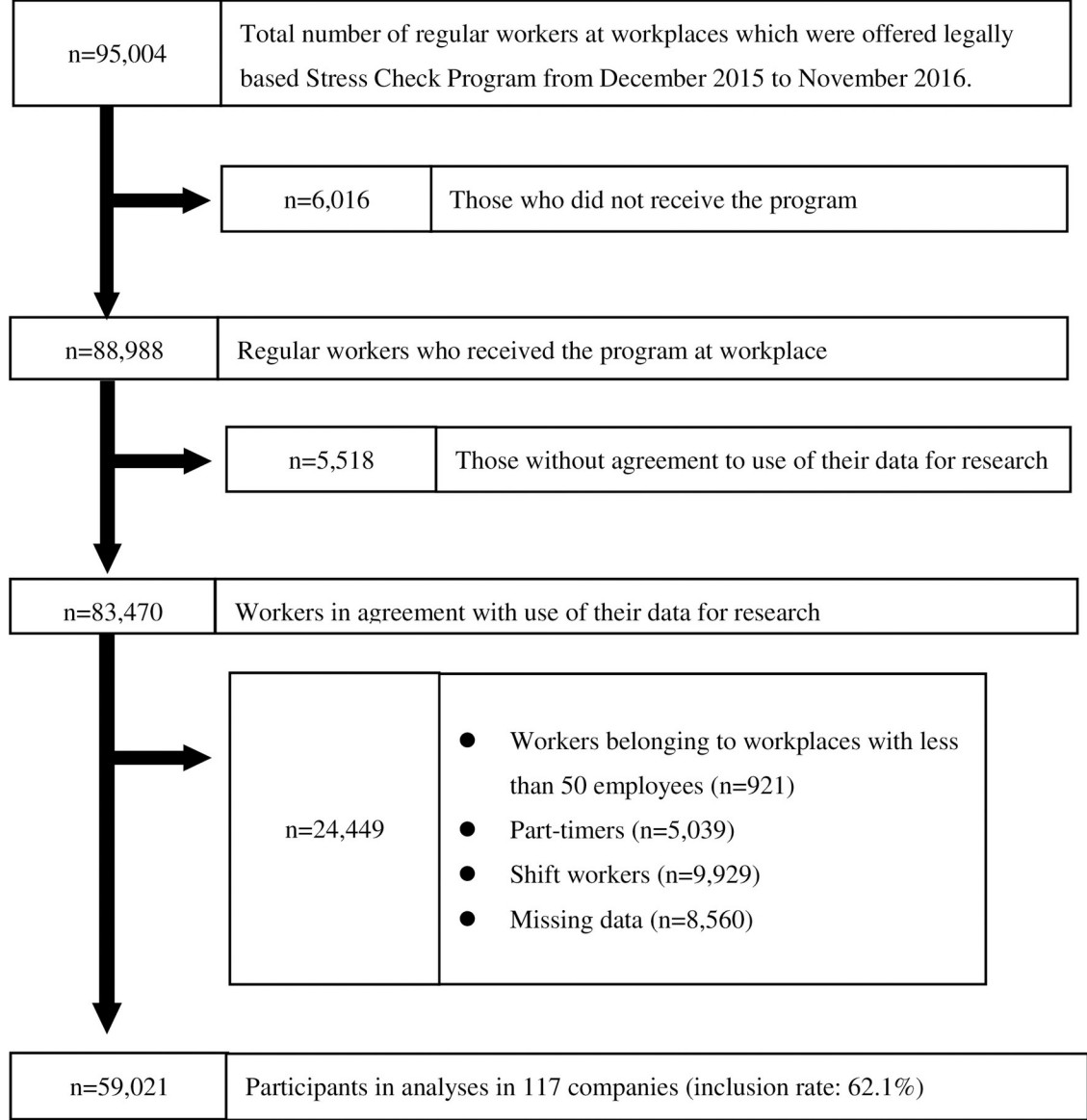

**Fig 1. Flow chart of study participants.**

for exclusion. Finally, 59,201 participants in 117 companies remained for this analysis (inclusion rate: 62.1%).

Table 1 shows the characteristics of study participants. The mean (±standard deviation) age was 44.3 (±10.7) years. Men accounted for 69.1% of the participants. The core group comprised middle-aged, male clerk with regular employment status and inflexible type of schedule. Average length of overtime was 26.3 (±20.5) hours per month. Regarding industries among participants, 32.9% (n = 19,401) and 25.7% (n = 15,150) of participants were workers in manufactural companies and public sectors, respectively. More than two thirds of participants were working in companies with 3,000 or more employees. Length of overtime differed significantly by gender, age, type of job, job class, employment status, type of schedule, industrial classification and company size.

**Table 1. Characteristics of participants.**

| | | Number of companies | Number of participants | (%) | Overtime work hours (hr/month) | | Proportion of excessive overtime work hours (%) | |
|---|---|---|---|---|---|---|---|---|
| | | | | | Mean | (S.D.) | 61–80 hr/month | 81-hr/month |
| Total | | 117 | 59,021 | (100.0%) | 26.3 | (±20.5) | (6.4%) | (1.5%) |
| Gender | | | | | | | | |
| | Men | | 40,764 | (69.1%) | 30.6 | (±21.9) | (8.7%) | (2.0%) |
| | Women | | 18,257 | (30.9%) | 16.6 | (±13.8) | (1.2%) | (0.5%) |
| Age | | | | | | | | |
| | <30 | | 7,518 | (12.7%) | 25.4 | (±20.0) | (4.8%) | (1.6%) |
| | 30–39 | | 13,400 | (22.7%) | 26.1 | (±20.1) | (5.4%) | (1.3%) |
| | 40–49 | | 20,238 | (34.3%) | 27.4 | (±21.1) | (7.0%) | (1.5%) |
| | 50–59 | | 14,671 | (24.9%) | 27.8 | (±21.8) | (8.4%) | (1.8%) |
| | ≥60 | | 3,194 | (5.4%) | 15.6 | (±14.3) | (1.5%) | (0.8%) |
| Types of job | | | | | | | | |
| | Sales | | 8,736 | (14.8%) | 41.9 | (±22.9) | (22.3%) | (1.5%) |
| | Professionals | | 15,401 | (26.1%) | 29.3 | (±21.4) | (5.7%) | (2.2%) |
| | Clerk | | 24,796 | (42.0%) | 20.8 | (±16.3) | (2.0%) | (0.9%) |
| | Service | | 3,143 | (5.3%) | 21.5 | (±18.3) | (4.5%) | (1.0%) |
| | Manual | | 2,649 | (4.5%) | 22.0 | (±19.3) | (4.5%) | (1.9%) |
| | Transport | | 530 | (0.9%) | 26.4 | (±26.6) | (5.1%) | (4.9%) |
| | Other | | 3,766 | (6.4%) | 21.0 | (±20.9) | (4.4%) | (2.5%) |
| Job class | | | | | | | | |
| | Regular | | 45,352 | (76.8%) | 23.5 | (±18.7) | (4.3%) | (0.9%) |
| | Manager | | 8,671 | (14.7%) | 41.2 | (±22.7) | (16.8%) | (3.5%) |
| | Director | | 2,294 | (3.9%) | 38.1 | (±26.5) | (14.7%) | (6.4%) |
| | Other | | 2,704 | (4.6%) | 15.3 | (±15.4) | (1.8%) | (1.1%) |
| Employment status | | | | | | | | |
| | Regular | | 45,427 | (77.0%) | 30.0 | (±21.6) | (8.2%) | (1.8%) |
| | Contract | | 10,979 | (18.6%) | 14.0 | (±10.6) | (0.3%) | (0.4%) |
| | Temporary | | 1,685 | (2.9%) | 11.6 | (±9.0) | (0.2%) | (0.4%) |
| | Other | | 930 | (1.6%) | 16.6 | (±15.5) | (0.9%) | (1.2%) |
| Types of schedule | | | | | | | | |
| | Inflexible | | 53,805 | (91.2%) | 26.5 | (±20.8) | (6.6%) | (1.4%) |
| | Flexible | | 4,442 | (7.5%) | 24.8 | (±20.3) | (4.4%) | (2.1%) |
| | Other | | 774 | (1.3%) | 18.4 | (±21.9) | (2.8%) | (3.2%) |
| Industrial Classification | | | | | | | | |
| | Construction | 5 | 9,399 | (15.9%) | 45.1 | (±19.7) | (19.7%) | (1.2%) |
| | Manufacture | 40 | 19,401 | (32.9%) | 25.4 | (±21.2) | (6.6%) | (2.1%) |
| | Electricity, gas, heat supply and water | 3 | 179 | (0.3%) | 40.5 | (±27.9) | (11.2%) | (8.4%) |
| | Information and communications | 6 | 2,137 | (3.6%) | 25.3 | (±16.7) | (3.7%) | (0.4%) |
| | Transport and postal activities | 7 | 387 | (0.7%) | 30.3 | (±29.2) | (8.0%) | (5.7%) |
| | Trade | 6 | 1,655 | (2.8%) | 28.1 | (±18.5) | (6.8%) | (0.4%) |
| | Finance | 2 | 491 | (0.8%) | 28.9 | (±17.8) | (5.1%) | (0.6%) |
| | Scientific and development research institutes | 3 | 183 | (0.3%) | 19.6 | (±16.4) | (2.2%) | (1.6%) |
| | Accommodations, eating and drinking services | 2 | 252 | (0.4%) | 30.6 | (±19.0) | (2.0%) | (3.2%) |

*(Continued)*

**Table 1.** (Continued)

|  |  | Number of companies | Number of participants | (%) | Overtime work hours (hr/month) | | Proportion of excessive overtime work hours (%) | |
|---|---|---|---|---|---|---|---|---|
|  |  |  |  |  | Mean | (S.D.) | 61–80 hr/month | 81-hr/month |
|  | Education | 3 | 961 | (1.6%) | 24.6 | (±26.3) | (4.2%) | (5.5%) |
|  | Medical service and social welfare | 17 | 2,593 | (4.4%) | 24.6 | (±25.7) | (3.5%) | (5.4%) |
|  | Others | 15 | 6,233 | (10.6%) | 14.3 | (±11.5) | (0.7%) | (0.6%) |
|  | Public sector | 8 | 15,150 | (25.7%) | 20.7 | (±14.7) | (1.3%) | (0.4%) |
| Company size (number of employees) |  |  |  |  |  |  |  |  |
|  | 50–99 | 42 | 1,848 | (3.1%) | 24.4 | (±23.0) | (4.1%) | (3.8%) |
|  | 100–299 | 44 | 4,363 | (7.4%) | 24.5 | (±21.5) | (4.9%) | (2.9%) |
|  | 300–999 | 18 | 5,108 | (8.7%) | 23.6 | (±19.0) | (4.6%) | (1.3%) |
|  | 1,000–2,999 | 8 | 7,448 | (12.6%) | 26.2 | (±21.5) | (4.9%) | (2.6%) |
|  | ≥3,000 | 5 | 40,254 | (68.2%) | 26.9 | (±20.7) | (7.2%) | (1.1%) |

Fig 2 shows the results of multiple linear regression analysis for total participants. Among those with 21–80 hours/month of overtime, linear dose-response curves were observed for "irritability", "fatigue", "anxiety", "depression" and "somatic responses", i.e. the more overtime a participant worked, the severer their stress response levels were. Particularly, a linear trend

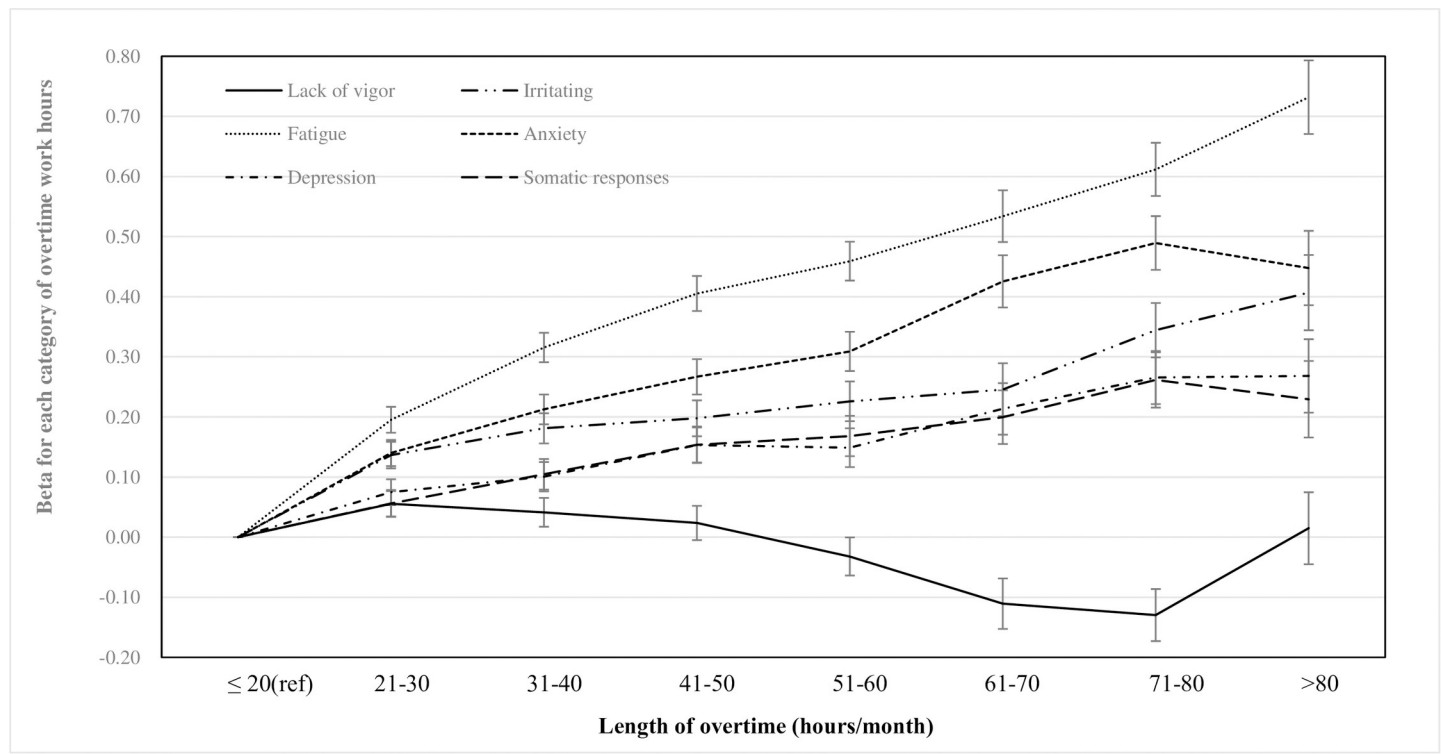

**Fig 2. Associations between stress response and overtime work hours; Multiple linear regression results.** * Higher score indicates unfavorable stress-response. Adjusted covariates are gender, age, type of job, job class, employment status, type of schedule, company size, company industry, job control, supervisor's support and coworker's support. ** Lack of vigor was derived by reversing the score of vigor for harmonization with other stress-response scales, which higher score indicates unfavorable stress-response.

was clearly found between "fatigue" level and overtime work hours. The beta (95% confidence intervals) of fatigue levels for each overtime category were 0.20 (0.17, 0.22), 0.32 (0.29, 0.34), 0.41 (0.38, 0.43), 0.46 (0.43, 0.49), 0.53 (0.49, 0.58), 0.61 (0.57, 0.66), 0.73 (0.67, 0.79), for, 21–30, 31–40, 41–50, 51–60, 61–70, 71–80 and 81 or more hours/month, respectively. Whereas, the dose-response curve for perceived "lack of vigor" was considerably different from those of other stress-responses. The "lack of vigor" levels remained were comparable across those with less than 60 hours of overtime work hours. Meanwhile, "lack of vigor" levels were lower among those with 61–80 hours of overtime (beta [95% CI]: -0.13 [-0.17, -0.08] and -0.11[-0.16, -0.06] for 61–70 and 71–80 hours/month, respectively). In other words, those with 61–80 hours of overtime were reported to be more vigorous at work, compared to those with less than 60 hours of overtime.

Table 2 shows the multiple linear regression results stratified by gender. For both genders, workers with 21 hours/month or more overtime reported significantly higher "irritability",

**Table 2. Associations between overtime work hours and stress responses: Gender-stratified multiple linear regression results.**

| Overtime work hours/month | n | Lack of Vigor | | | Irritability | | | Fatigue | | | Anxiety | | | Depression | | | Somatic responses | | |
|---|---|---|---|---|---|---|---|---|---|---|---|---|---|---|---|---|---|---|---|
| | | beta | 95% CI | p | beta | 95% CI | p | beta | 95% CI | p | beta | 95% CI | p | beta | 95% CI | p | beta | 95% CI | p |
| **Men** | | | | | | | | | | | | | | | | | | | |
| ≤20 | 15,751 | Ref | | | Ref | | | Ref | | | Ref | | | Ref | | | Ref | | |
| 21–30 | 7,260 | 0.04 | (0.02, 0.07) | ** | 0.13 | (0.10, 0.16) | *** | 0.16 | (0.13, 0.18) | *** | 0.11 | (0.08, 0.14) | *** | 0.04 | (0.02, 0.07) | ** | 0.03 | (0.00, 0.05) | * |
| 31–40 | 5,812 | 0.01 | (-0.02, 0.04) | | 0.17 | (0.14, 0.19) | *** | 0.26 | (0.23, 0.29) | *** | 0.16 | (0.13, 0.19) | *** | 0.06 | (0.03, 0.08) | *** | 0.07 | (0.04, 0.10) | *** |
| 41–50 | 4,097 | 0.00 | (-0.03, 0.03) | | 0.19 | (0.16, 0.22) | *** | 0.36 | (0.33, 0.39) | *** | 0.23 | (0.20, 0.26) | *** | 0.12 | (0.09, 0.15) | *** | 0.13 | (0.10, 0.17) | *** |
| 51–60 | 3,494 | -0.04 | (-0.07, 0.00) | * | 0.22 | (0.18, 0.25) | *** | 0.42 | (0.38, 0.45) | *** | 0.26 | (0.23, 0.29) | *** | 0.10 | (0.07, 0.13) | *** | 0.14 | (0.10, 0.17) | *** |
| 61–70 | 1,814 | -0.13 | (-0.17, -0.09) | *** | 0.23 | (0.19, 0.28) | *** | 0.48 | (0.44, 0.53) | *** | 0.38 | (0.33, 0.42) | *** | 0.16 | (0.12, 0.21) | *** | 0.16 | (0.11, 0.21) | *** |
| 71–80 | 1,740 | -0.13 | (-0.18, -0.09) | *** | 0.33 | (0.28, 0.38) | *** | 0.57 | (0.52, 0.61) | *** | 0.44 | (0.40, 0.49) | *** | 0.22 | (0.17, 0.26) | *** | 0.23 | (0.18, 0.27) | *** |
| 81- | 796 | -0.01 | (-0.07, 0.06) | | 0.41 | (0.35, 0.48) | *** | 0.73 | (0.66, 0.79) | *** | 0.42 | (0.35, 0.48) | *** | 0.24 | (0.17, 0.30) | *** | 0.23 | (0.16, 0.29) | *** |
| p for linear trend | | <0.001 (negative) | | | <0.001 (positive) | | | <0.001 (positive) | | | <0.001 (positive) | | | <0.001 (positive) | | | <0.001 (positive) | | |
| **Women** | | | | | | | | | | | | | | | | | | | |
| ≤20 | 13,529 | Ref | | | Ref | | | Ref | | | Ref | | | Ref | | | Ref | | |
| 21–30 | 2,301 | 0.07 | (0.03, 0.11) | *** | 0.15 | (0.10, 0.19) | *** | 0.26 | (0.21, 0.30) | *** | 0.19 | (0.14, 0.23) | *** | 0.13 | (0.09, 0.17) | *** | 0.11 | (0.07, 0.16) | *** |
| 31–40 | 1,219 | 0.15 | (0.10, 0.20) | *** | 0.23 | (0.17, 0.28) | *** | 0.47 | (0.21, 0.31) | *** | 0.36 | (0.31, 0.42) | *** | 0.20 | (0.15, 0.25) | *** | 0.19 | (0.13, 0.25) | *** |
| 41–50 | 567 | 0.14 | (0.06, 0.22) | *** | 0.20 | (0.12, 0.29) | *** | 0.52 | (0.21, 0.32) | *** | 0.34 | (0.26, 0.42) | *** | 0.22 | (0.14, 0.30) | *** | 0.18 | (0.10, 0.26) | *** |
| 51–60 | 326 | -0.04 | (-0.14, 0.07) | | 0.21 | (0.11, 0.32) | *** | 0.60 | (0.21, 0.33) | *** | 0.50 | (0.39, 0.61) | *** | 0.33 | (0.23, 0.43) | *** | 0.25 | (0.14, 0.36) | *** |
| 61–70 | 127 | 0.12 | (-0.04, 0.28) | | 0.25 | (0.08, 0.42) | ** | 0.82 | (0.21, 0.34) | *** | 0.71 | (0.54, 0.88) | *** | 0.54 | (0.38, 0.70) | *** | 0.41 | (0.24, 0.58) | *** |
| 71–80 | 97 | -0.14 | (-0.32, 0.05) | | 0.39 | (0.19, 0.58) | *** | 0.78 | (0.21, 0.35) | *** | 0.71 | (0.52, 0.90) | *** | 0.60 | (0.42, 0.79) | *** | 0.44 | (0.24, 0.63) | *** |
| 81- | 91 | 0.01 | (-0.18, 0.20) | | 0.32 | (0.12, 0.52) | ** | 0.62 | (0.21, 0.36) | *** | 0.53 | (0.33, 0.72) | *** | 0.34 | (0.15, 0.52) | ** | 0.13 | (-0.07, 0.33) | |
| p for linear trend | | <0.001 (positive) | | | <0.001 (positive) | | | <0.001 (positive) | | | <0.001 (positive) | | | <0.001 (positive) | | | <0.001 (positive) | | |
| **Interaction terms** | | | | | | | | | | | | | | | | | | | |

(*Continued*)

Table 2. (Continued)

| Overtime work hours/month | n | Lack of Vigor | | | Irritability | | | Fatigue | | | Anxiety | | | Depression | | | Somatic responses | | |
|---|---|---|---|---|---|---|---|---|---|---|---|---|---|---|---|---|---|---|---|
| | | beta | 95% CI | p | beta | 95% CI | p | beta | 95% CI | p | beta | 95% CI | p | beta | 95% CI | p | beta | 95% CI | p |
| **Men (ref) versus Women** | | | | | | | | | | | | | | | | | | | |
| 21–30 | | 0.04 | (-0.01, 0.08) | | 0.02 | (-0.03, 0.06) | | 0.09 | (0.04, 0.13) | *** | 0.07 | (0.02, 0.11) | ** | 0.08 | (0.03, 0.13) | ** | 0.09 | (0.04, 0.14) | *** |
| 31–40 | | 0.15 | (0.09, 0.21) | *** | 0.06 | (0.00, 0.12) | + | 0.19 | (0.13, 0.25) | *** | 0.18 | (0.12, 0.24) | *** | 0.13 | (0.07, 0.19) | *** | 0.12 | (0.06, 0.19) | *** |
| 41–50 | | 0.15 | (0.06, 0.23) | *** | 0.01 | (-0.08, 0.09) | | 0.13 | (0.05, 0.22) | ** | 0.08 | (0.00, 0.16) | + | 0.08 | (0.00, 0.17) | * | 0.04 | (-0.04, 0.13) | |
| 51–60 | | 0.04 | (-0.06, 0.14) | | -0.03 | (-0.14, 0.08) | | 0.13 | (0.03, 0.24) | * | 0.18 | (0.08, 0.29) | ** | 0.20 | (0.09, 0.30) | *** | 0.09 | (-0.02, 0.20) | |
| 61–70 | | 0.29 | (0.12, 0.45) | ** | -0.02 | (-0.19, 0.15) | | 0.30 | (0.13, 0.46) | *** | 0.28 | (0.12, 0.45) | ** | 0.35 | (0.18, 0.51) | *** | 0.21 | (0.04, 0.38) | * |
| 71–80 | | 0.03 | (-0.15, 0.22) | | 0.01 | (-0.18, 0.21) | | 0.16 | (-0.03, 0.35) | + | 0.20 | (0.01, 0.39) | * | 0.34 | (0.16, 0.53) | *** | 0.17 | (-0.03, 0.36) | + |
| 81- | | -0.01 | (-0.20, 0.19) | | -0.09 | (-0.29, 0.12) | | -0.09 | (-0.29, 0.11) | | 0.10 | (-0.10, 0.30) | | 0.11 | (-0.09, 0.31) | | -0.08 | (-0.29, 0.12) | |

+ p<0.1

*p<0.05, ** p<0.01, ***p<0.001, Higher score indicates unfavorable stress-response. All betas were adjusted by age, type of job, job class, employment status, type of schedule, company size, company industry, job control, supervisors support and coworker's support. Lack of vigor was derived by reversing the score of vigor for harmonization with other stress-response scales, which higher score indicates unfavorable stress-response

"fatigue", "anxiety", "depression" and "somatic responses", compared to those with less than 20 hours/month of overtime (p<0.001). In addition, workers who engaged in 21–30 hours/month of overtime reported a higher level of "lack of vigor". However, male workers who engaged in 61–80 hours/month of overtime reported a significantly lower level of "lack of vigor" than those with less than 20 hours/month of overtime (51–60 hours/month: beta = -0.04 [95%CI: -0.07, 0.00], 61–70 hours/month: beta = -0.13 [-0.17. -0.09], 71–80 hours/month: beta = -0.13 [-0.18, -0.09]). This association was not observed for women. Both among men and women there is no increase in effect above 80 hours overtime work/month. Indeed the data did not include any clinical data, however speculatively, there might be a healthy worker effect in this extreme group. That would mean that unhealthy workers in this category either left a job or stopped working whereas healthy workers would stay. This may lead to underestimation of effect. Then, linear trend tests showed that length of overtime was positively associated with "irritability", "fatigue", "anxiety" "depression" and "somatic responses" (p for trend <0.001). Conversely, length of overtime was negatively associated with "lack of vigor" (p for trend <0.001). None of these results became significantly different after using the dataset with multiple imputation. For gender differences, significant interactions by gender were observed in the association between overtime work hours and several stress responses. Compared to workers with shorter overtime work hours, those with longer hours tend to have higher stress responses for both genders; however, this association was stronger among women than men. This implies that working women may be more vulnerable than men for long working hours.

Tables 3 and 4 showed stratified analysis results by age-category and job-class. Compared to middle-aged male workers, older workers showed relatively higher stress responses, especially "fatigue", "anxiety" and "depression". This may suggest that older workers constitute a more vulnerable population than younger ones for overtime working than middle-aged

**Table 3. Association between overtime work hours and stress responses: Results of multiple linear regression stratified by gender and age-category.**

| Overtime work hours/month | n | Lack of Vigor | | | Irritability | | | Fatigue | | | Anxiety | | | Depression | | | Somatic responses | | |
|---|---|---|---|---|---|---|---|---|---|---|---|---|---|---|---|---|---|---|---|
| | | beta | 95% CI | p | beta | 95% CI | p | beta | 95% CI | p | beta | 95% CI | p | beta | 95% CI | p | beta | 95% CI | p |
| **Men** | | | | | | | | | | | | | | | | | | | |
| **Younger (aged less than 30 years)** | | | | | | | | | | | | | | | | | | | |
| ≤20 | 1,485 | Ref | | | Ref | | | Ref | | | Ref | | | Ref | | | Ref | | |
| 21–30 | 796 | 0.12 | (0.04, 0.20) | ** | 0.15 | (0.07, 0.24) | *** | 0.22 | (0.14, 0.30) | *** | 0.12 | (0.03, 0.20) | ** | 0.12 | (0.04, 0.21) | ** | 0.03 | (-0.06, 0.12) | |
| 31–40 | 546 | 0.08 | (-0.01, 0.17) | + | 0.17 | (0.07, 0.27) | ** | 0.28 | (0.19, 0.37) | *** | 0.15 | (0.05, 0.24) | ** | 0.06 | (-0.03, 0.16) | | 0.10 | (0.00, 0.20) | |
| 41–50 | 345 | 0.08 | (-0.03, 0.18) | | 0.21 | (0.09, 0.33) | ** | 0.45 | (0.35, 0.56) | *** | 0.30 | (0.19, 0.42) | *** | 0.28 | (0.17, 0.40) | *** | 0.32 | (0.20, 0.44) | *** |
| 51–60 | 291 | -0.05 | (-0.17, 0.07) | | 0.29 | (0.16, 0.42) | *** | 0.48 | (0.36, 0.59) | *** | 0.28 | (0.15, 0.41) | *** | 0.14 | (0.01, 0.26) | * | 0.15 | (0.01, 0.28) | * |
| 61–70 | 146 | -0.01 | (-0.16, 0.14) | | 0.35 | (0.18, 0.52) | *** | 0.57 | (0.42, 0.72) | *** | 0.49 | (0.33, 0.66) | *** | 0.37 | (0.20, 0.54) | ** | 0.30 | (0.13, 0.48) | *** |
| 71–80 | 135 | -0.12 | (-0.28, 0.04) | | 0.40 | (0.22, 0.58) | *** | 0.42 | (0.26, 0.58) | *** | 0.51 | (0.34, 0.69) | *** | 0.41 | (0.23, 0.59) | *** | 0.38 | (0.20, 0.56) | *** |
| 81- | 77 | 0.05 | (-0.16, 0.25) | | 0.48 | (0.26, 0.71) | *** | 0.89 | (0.69, 1.09) | *** | 0.44 | (0.22, 0.66) | *** | 0.44 | (0.22, 0.66) | ** | 0.27 | (0.04, 0.50) | * |
| p for linear trend | | <0.001 (negative) | | | <0.001 (positive) | | | <0.001 (positive) | | | <0.001 (positive) | | | <0.001 (positive) | | | <0.001 (positive) | | |
| **Middle (aged 30–59 years)** | | | | | | | | | | | | | | | | | | | |
| ≤20 | 12,097 | Ref | | | Ref | | | Ref | | | Ref | | | Ref | | | Ref | | |
| 21–30 | 6,132 | 0.01 | (-0.02, 0.03) | | 0.11 | (0.08, 0.14) | *** | 0.13 | (0.10, 0.16) | *** | 0.08 | (0.05, 0.11) | *** | 0.01 | (-0.02, 0.04) | | 0.01 | (-0.02, 0.04) | |
| 31–40 | 5,038 | -0.03 | (-0.06, 0.00) | * | 0.14 | (0.11, 0.17) | *** | 0.24 | (0.21, 0.27) | *** | 0.13 | (0.10, 0.16) | *** | 0.03 | (0.00, 0.06) | + | 0.04 | (0.01, 0.07) | ** |
| 41–50 | 3,624 | -0.04 | (-0.07, 0.00) | * | 0.16 | (0.13, 0.20) | *** | 0.33 | (0.30, 0.36) | *** | 0.19 | (0.15, 0.22) | *** | 0.08 | (0.04, 0.11) | *** | 0.09 | (0.06, 0.13) | *** |
| 51–60 | 3,129 | -0.07 | (-0.10, -0.03) | *** | 0.19 | (0.15, 0.22) | *** | 0.39 | (0.35, 0.42) | *** | 0.22 | (0.19, 0.26) | *** | 0.07 | (0.04, 0.11) | *** | 0.11 | (0.08, 0.15) | *** |
| 61–70 | 1,617 | -0.17 | (-0.21, -0.12) | *** | 0.20 | (0.16, 0.25) | *** | 0.45 | (0.41, 0.50) | *** | 0.33 | (0.28, 0.37) | *** | 0.12 | (0.07, 0.17) | *** | 0.13 | (0.08, 0.18) | *** |
| 71–80 | 1,577 | -0.17 | (-0.22, -0.12) | *** | 0.30 | (0.25, 0.35) | *** | 0.56 | (0.51, 0.61) | *** | 0.40 | (0.35, 0.45) | *** | 0.18 | (0.13, 0.22) | *** | 0.19 | (0.14, 0.24) | *** |
| 81- | 683 | -0.04 | (-0.10, 0.03) | | 0.40 | (0.33, 0.47) | *** | 0.70 | (0.63, 0.77) | *** | 0.40 | (0.33, 0.47) | *** | 0.20 | (0.13, 0.27) | *** | 0.20 | (0.13, 0.27) | *** |
| p for linear trend | | <0.001 (negative) | | | <0.001 (positive) | | | <0.001 (positive) | | | <0.001 (positive) | | | <0.001 (positive) | | | <0.001 (positive) | | |
| **Older (aged 60 or older)** | | | | | | | | | | | | | | | | | | | |
| ≤20 | 2,169 | Ref | | | Ref | | | Ref | | | Ref | | | Ref | | | Ref | | |
| 21–30 | 332 | 0.01 | (-0.09, 0.12) | | 0.08 | (-0.02, 0.18) | | 0.24 | (0.15, 0.33) | *** | 0.23 | (0.13, 0.32) | *** | 0.13 | (0.03, 0.22) | ** | 0.07 | (-0.04, 0.17) | |
| 31–40 | 228 | 0.09 | (-0.03, 0.22) | | 0.29 | (0.17, 0.41) | *** | 0.34 | (0.23, 0.45) | *** | 0.31 | (0.20, 0.42) | *** | 0.23 | (0.12, 0.34) | *** | 0.22 | (0.09, 0.34) | ** |
| 41–50 | 128 | -0.06 | (-0.22, 0.10) | | 0.26 | (0.10, 0.41) | ** | 0.52 | (0.37, 0.66) | *** | 0.43 | (0.28, 0.57) | *** | 0.30 | (0.16, 0.45) | *** | 0.30 | (0.13, 0.46) | *** |
| 51–60 | 74 | -0.04 | (-0.25, 0.17) | | 0.27 | (0.06, 0.46) | * | 0.57 | (0.38, 0.75) | *** | 0.48 | (0.29, 0.66) | *** | 0.30 | (0.11, 0.48) | ** | 0.37 | (0.16, 0.58) | ** |
| 61–70 | 51 | -0.36 | (-0.61, -0.11) | ** | 0.15 | (-0.09, 0.39) | | 0.50 | (0.28, 0.72) | *** | 0.51 | (0.29, 0.73) | *** | 0.20 | (-0.02, 0.42) | + | 0.20 | (-0.05, 0.45) | |
| 71–80 | 28 | 0.14 | (-0.20, 0.47) | | 0.40 | (0.08, 0.72) | * | 0.68 | (0.38, 0.97) | *** | 0.73 | (0.44, 1.03) | *** | 0.31 | (0.02, 0.61) | * | 0.47 | (0.14, 0.81) | ** |
| 81- | 36 | 0.05 | (-0.25, 0.35) | | 0.10 | (-0.18, 0.38) | | 0.59 | (0.33, 0.85) | *** | 0.32 | (0.06, 0.58) | ** | 0.16 | (-0.11, 0.42) | | 0.29 | (-0.01, 0.58) | + |
| p for linear trend | | n.s. | | | <0.001 (positive) | | | <0.001 (positive) | | | <0.001 (positive) | | | <0.001 (positive) | | | <0.001 (positive) | | |

(*Continued*)

**Table 3.** (Continued)

| Overtime work hours/month | n | Lack of Vigor | | | Irritability | | | Fatigue | | | Anxiety | | | Depression | | | Somatic responses | | |
|---|---|---|---|---|---|---|---|---|---|---|---|---|---|---|---|---|---|---|---|
| | | beta | 95% CI | p | beta | 95% CI | p | beta | 95% CI | p | beta | 95% CI | p | beta | 95% CI | p | beta | 95% CI | p |
| **Interaction between overtime and age-category** | | | | | | | | | | | | | | | | | | | |
| **Middle (ref) versus Young** | | | | | | | | | | | | | | | | | | | |
| 21–30 | | 0.14 | (0.05, 0.21) | ** | 0.04 | (-0.04, 0.13) | | 0.06 | (-0.02, 0.15) | | 0.03 | (-0.05, 0.11) | | 0.11 | (0.03, 0.19) | * | 0.04 | (-0.05, 0.13) | |
| 31–40 | | 0.16 | (0.07, 0.25) | ** | 0.04 | (-0.06, 0.13) | | 0.03 | (-0.07, 0.12) | | 0.01 | (-0.08, 0.10) | | 0.04 | (-0.06, 0.13) | | 0.08 | (-0.02, 0.17) | |
| 41–50 | | 0.16 | (0.05, 0.27) | ** | 0.06 | (-0.05, 0.17) | | 0.10 | (-0.01, 0.21) | | 0.10 | (-0.02, 0.21) | | 0.21 | (0.10, 0.32) | *** | 0.24 | (0.13, 0.36) | *** |
| 51–60 | | 0.09 | (-0.02, 0.21) | | 0.12 | (0.00, 0.24) | * | 0.07 | (-0.05, 0.18) | | 0.04 | (-0.08, 0.15) | | 0.08 | (-0.04, 0.20) | | 0.06 | (-0.06, 0.18) | |
| 61–70 | | 0.24 | (0.08, 0.39) | ** | 0.18 | (0.01, 0.34) | * | 0.13 | (-0.03, 0.29) | | 0.17 | (0.01, 0.33) | * | 0.30 | (0.14, 0.46) | *** | 0.20 | (0.03, 0.37) | * |
| 71–80 | | 0.13 | (-0.03, 0.29) | | 0.12 | (-0.05, 0.28) | | -0.14 | (-0.30, 0.02) | + | 0.11 | (-0.06, 0.27) | | 0.27 | (0.10, 0.44) | ** | 0.22 | (0.05, 0.39) | * |
| 81- | | 0.08 | (-0.13, 0.29) | | 0.08 | (-0.14, 0.30) | | 0.14 | (-0.07, 0.36) | | 0.01 | (-0.21, 0.22) | | 0.19 | (-0.03, 0.41) | + | 0.06 | (-0.16, 0.29) | |
| **Middle (ref) versus Older** | | | | | | | | | | | | | | | | | | | |
| 21–30 | | 0.06 | (-0.04, 0.17) | | -0.01 | (-0.12, 0.10) | | 0.12 | (0.02, 0.23) | * | 0.16 | (0.05, 0.27) | ** | 0.14 | (0.03, 0.24) | * | 0.10 | (-0.01, 0.21) | + |
| 31–40 | | 0.21 | (0.09, 0.33) | ** | 0.20 | (0.07, 0.33) | * | 0.14 | (0.01, 0.26) | * | 0.19 | (0.07, 0.32) | ** | 0.22 | (0.10, 0.35) | ** | 0.23 | (0.10, 0.36) | *** |
| 41–50 | | 0.09 | (-0.07, 0.25) | | 0.17 | (0.00, 0.33) | + | 0.25 | (0.09, 0.41) | * | 0.25 | (0.09, 0.42) | ** | 0.27 | (0.10, 0.43) | ** | 0.27 | (0.10, 0.44) | ** |
| 51–60 | | 0.15 | (-0.05, 0.36) | | 0.16 | (-0.05, 0.38) | | 0.24 | (0.03, 0.45) | * | 0.25 | (0.04, 0.46) | * | 0.27 | (0.05, 0.48) | * | 0.33 | (0.11, 0.55) | ** |
| 61–70 | | -0.08 | (-0.33, 0.17) | | 0.04 | (-0.22, 0.29) | | 0.12 | (-0.13, 0.38) | | 0.20 | (-0.06, 0.45) | | 0.14 | (-0.12, 0.40) | | 0.14 | (-0.12, 0.41) | |
| 71–80 | | 0.40 | (0.07, 0.73) | * | 0.18 | (-0.16, 0.53) | | 0.18 | (-0.15, 0.51) | | 0.35 | (0.01, 0.69) | * | 0.21 | (-0.13, 0.55) | | 0.36 | (0.01, 0.72) | * |
| 81- | | 0.09 | (-0.20, 0.39) | | -0.30 | (-0.61, 0.01) | + | -0.15 | (-0.45, 0.16) | | -0.06 | (-0.36, 0.25) | | -0.02 | (-0.32, 0.29) | | 0.08 | (-0.23, 0.40) | |
| **Women** | | | | | | | | | | | | | | | | | | | |
| **Younger (aged less than 30 years)** | | | | | | | | | | | | | | | | | | | |
| ≤20 | 1,665 | Ref | | | Ref | | | Ref | | | Ref | | | Ref | | | Ref | | |
| 21–30 | 473 | 0.02 | (-0.07, 0.11) | | 0.13 | (0.02, 0.23) | * | 0.23 | (0.14, 0.33) | *** | 0.09 | (-0.02, 0.19) | + | 0.10 | (0.00, 0.20) | * | 0.08 | (-0.02, 0.18) | |
| 31–40 | 246 | 0.14 | (0.01, 0.26) | * | 0.18 | (0.04, 0.32) | * | 0.37 | (0.24, 0.49) | *** | 0.25 | (0.12, 0.38) | *** | 0.26 | (0.13, 0.39) | *** | 0.06 | (-0.08, 0.20) | |
| 41–50 | 96 | 0.11 | (-0.08, 0.29) | | 0.10 | (-0.11, 0.31) | | 0.46 | (0.26, 0.65) | *** | 0.34 | (0.13, 0.54) | ** | 0.23 | (0.03, 0.43) | * | 0.09 | (-0.12, 0.30) | |
| 51–60 | 81 | 0.11 | (-0.09, 0.32) | | 0.11 | (-0.12, 0.34) | | 0.66 | (0.44, 0.87) | *** | 0.47 | (0.25, 0.69) | *** | 0.47 | (0.25, 0.69) | *** | 0.24 | (0.01, 0.47) | * |
| 61–70 | 15 | 0.45 | (-0.01, 0.91) | | 0.26 | (-0.26, 0.79) | | 0.58 | (0.10, 1.06) | * | 0.22 | (-0.28, 0.72) | | 0.49 | (0.00, 0.98) | + | 0.31 | (-0.21, 0.83) | |
| 71–80 | 16 | -0.24 | (-0.67, 0.20) | | 0.53 | (0.02, 1.03) | * | 0.78 | (0.32, 1.25) | ** | 0.73 | (0.24, 1.21) | ** | 0.69 | (0.22, 1.16) | ** | 0.35 | (-0.15, 0.85) | |
| 81- | 20 | -0.06 | (-0.46, 0.33) | | 0.06 | (-0.39, 0.51) | | 0.54 | (0.12, 0.95) | * | 0.24 | (-0.20, 0.67) | | 0.37 | (-0.05, 0.80) | + | 0.06 | (-0.38, 0.51) | |
| p for linear trend | | <0.001 (positive) | | | <0.001 (positive) | | | <0.001 (positive) | | | <0.001 (positive) | | | <0.001 (positive) | | | <0.001 (positive) | | |
| **Middle (aged 30–59 years)** | | | | | | | | | | | | | | | | | | | |
| ≤20 | 10,896 | Ref | | | Ref | | | Ref | | | Ref | | | Ref | | | Ref | | |

(*Continued*)

**Table 3.** (Continued)

| Overtime work hours/month | n | Lack of Vigor | | | Irritability | | | Fatigue | | | Anxiety | | | Depression | | | Somatic responses | | |
|---|---|---|---|---|---|---|---|---|---|---|---|---|---|---|---|---|---|---|---|
| | | beta | 95% CI | p | beta | 95% CI | p | beta | 95% CI | p | beta | 95% CI | p | beta | 95% CI | p | beta | 95% CI | p |
| 21–30 | 1,735 | 0.08 | (0.03, 0.13) | ** | 0.15 | (0.10, 0.20) | *** | 0.27 | (0.22, 0.32) | *** | 0.20 | (0.15, 0.25) | *** | 0.13 | (0.08, 0.17) | *** | 0.12 | (0.07, 0.17) | *** |
| 31–40 | 942 | 0.16 | (0.10, 0.22) | *** | 0.24 | (0.18, 0.30) | *** | 0.50 | (0.44, 0.57) | *** | 0.39 | (0.32, 0.45) | *** | 0.19 | (0.12, 0.25) | *** | 0.22 | (0.16, 0.29) | *** |
| 41–50 | 462 | 0.13 | (0.05, 0.22) | ** | 0.22 | (0.12, 0.30) | *** | 0.53 | (0.44, 0.62) | *** | 0.33 | (0.24, 0.42) | *** | 0.21 | (0.13, 0.29) | *** | 0.18 | (0.09, 0.27) | *** |
| 51–60 | 240 | -0.12 | (-0.23, 0.00) | + | 0.23 | (0.10, 0.35) | *** | 0.57 | (0.44, 0.70) | *** | 0.49 | (0.37, 0.62) | *** | 0.26 | (0.14, 0.38) | *** | 0.24 | (0.12, 0.37) | *** |
| 61–70 | 110 | 0.05 | (-0.13, 0.22) | | 0.22 | (0.04, 0.40) | * | 0.84 | (0.65, 1.02) | *** | 0.77 | (0.59, 0.95) | *** | 0.54 | (0.37, 0.71) | *** | 0.41 | (0.23, 0.59) | *** |
| 71–80 | 81 | -0.13 | (-0.33, 0.07) | | 0.34 | (0.13, 0.55) | ** | 0.77 | (0.56, 0.98) | *** | 0.68 | (0.47, 0.89) | *** | 0.56 | (0.36, 0.76) | *** | 0.44 | (0.23, 0.65) | *** |
| 81- | 69 | 0.05 | (-0.17, 0.26) | | 0.41 | (0.18, 0.64) | *** | 0.64 | (0.41, 0.87) | *** | 0.61 | (0.38, 0.84) | ** | 0.34 | (0.13, 0.56) | ** | 0.13 | (-0.10, 0.36) | |
| p for linear trend | | <0.001 (positive) | | | <0.001 (positive) | | | <0.001 (positive) | | | <0.001 (positive) | | | <0.001 (positive) | | | <0.001 (positive) | | |
| **Older (aged 60 or older)** | | | | | | | | | | | | | | | | | | | |
| ≤20 | 968 | Ref | | | Ref | | | Ref | | | Ref | | | Ref | | | Ref | | |
| 21–30 | 93 | 0.17 | (-0.04, 0.39) | | 0.22 | (0.03, 0.41) | * | 0.27 | (0.08, 0.46) | ** | 0.29 | (0.11, 0.47) | ** | 0.23 | (0.06, 0.39) | ** | 0.08 | (-0.12, 0.28) | |
| 31–40 | 31 | -0.16 | (-0.52, 0.20) | | 0.21 | (-0.11, 0.53) | | 0.23 | (-0.09, 0.54) | | 0.21 | (-0.09, 0.51) | | 0.08 | (-0.20, 0.36) | | 0.09 | (-0.24, 0.41) | |
| 41–50 | 9 | 0.09 | (-0.56, 0.75) | | 0.35 | (-0.24, 0.93) | | 0.31 | (-0.27, 0.89) | | 0.67 | (0.12, 1.21) | * | 0.45 | (-0.06, 0.97) | + | -0.02 | (-0.62, 0.58) | |
| 51–60 | 5 | 0.99 | (0.12, 1.87) | * | 1.41 | (0.63, 2.19) | *** | 1.02 | (0.24, 1.80) | * | 0.71 | (-0.02, 1.44) | + | 1.08 | (0.39, 1.77) | ** | 0.35 | (-0.45, 1.16) | |
| 61–70 | 2 | 1.36 | (-0.02, 2.74) | + | 1.10 | (-0.13, 2.33) | + | 1.16 | (-0.07, 2.38) | | 0.56 | (-0.60, 1.71) | | 0.51 | (-0.58, 1.60) | | 0.21 | (-1.06, 1.48) | |
| 71–80 | 0 | - | | | - | | | - | | | - | | | - | | | - | | |
| 81- | 2 | -0.23 | (-1.61, 1.16) | | -0.37 | (-1.61, 0.86) | | 0.82 | (-0.41, 2.05) | | -0.18 | (-1.34, 0.98) | | -0.66 | (-1.75, 0.44) | | 0.32 | (-0.96, 1.60) | |
| p for linear trend | | n.s. | | | <0.001 (positive) | | | <0.001 (positive) | | | <0.001 (positive) | | | <0.01 (positive) | | | n.s. | | |
| **Interaction between overtime and age-category** | | | | | | | | | | | | | | | | | | | |
| **Middle (ref) versus Young** | | | | | | | | | | | | | | | | | | | |
| 21–30 | | -0.05 | (-0.15, 0.06) | | -0.02 | (-0.13, 0.09) | | -0.05 | (-0.16, 0.07) | | -0.14 | (-0.25, -0.03) | * | -0.03 | (-0.13, 0.07) | | -0.03 | (-0.14, 0.08) | * |
| 31–40 | | -0.01 | (-0.15, 0.12) | | -0.08 | (-0.23, 0.06) | | -0.15 | (-0.29, 0.00) | * | -0.16 | (-0.30, -0.01) | * | 0.06 | (-0.08, 0.20) | | -0.15 | (-0.29, 0.00) | |
| 41–50 | | 0.00 | (-0.20, 0.21) | | -0.14 | (-0.36, 0.08) | | -0.07 | (-0.29, 0.15) | | -0.02 | (-0.24, 0.20) | | 0.01 | (-0.20, 0.21) | | -0.06 | (-0.28, 0.16) | |
| 51–60 | | 0.27 | (0.04, 0.51) | * | -0.15 | (-0.40, 0.10) | | 0.09 | (-0.16, 0.34) | | -0.06 | (-0.30, 0.19) | | 0.20 | (-0.03, 0.44) | + | 0.05 | (-0.20, 0.29) | |
| 61–70 | | 0.39 | (-0.11, 0.89) | | 0.00 | (-0.52, 0.53) | | -0.24 | (-0.76, 0.29) | | -0.54 | (-1.06, -0.03) | * | -0.11 | (-0.60, 0.39) | | -0.05 | (-0.58, 0.47) | |
| 71–80 | | -0.07 | (-0.56, 0.42) | | 0.18 | (-0.34, 0.70) | | 0.06 | (-0.47, 0.58) | | 0.09 | (-0.42, 0.61) | | 0.18 | (-0.31, 0.67) | | -0.04 | (-0.56, 0.49) | |
| 81- | | -0.12 | (-0.58, 0.34) | | -0.35 | (-0.84, 0.13) | | -0.09 | (-0.58, 0.39) | | -0.35 | (-0.83, 0.12) | | 0.03 | (-0.43, 0.48) | | -0.04 | (-0.52, 0.45) | |
| **Middle (ref) versus Older** | | | | | | | | | | | | | | | | | | | |
| 21–30 | | 0.12 | (-0.08, 0.32) | | 0.10 | (-0.11, 0.31) | | 0.06 | (-0.15, 0.27) | | 0.13 | (-0.08, 0.33) | | 0.14 | (-0.06, 0.33) | | 0.01 | (-0.20, 0.22) | |

*(Continued)*

**Table 3.** (Continued)

| Overtime work hours/month | n | Lack of Vigor | | | Irritability | | | Fatigue | | | Anxiety | | | Depression | | | Somatic responses | | |
|---|---|---|---|---|---|---|---|---|---|---|---|---|---|---|---|---|---|---|---|
| | | beta | 95% CI | p | beta | 95% CI | p | beta | 95% CI | p | beta | 95% CI | p | beta | 95% CI | p | beta | 95% CI | p |
| 31–40 | | -0.27 | (-0.60, 0.06) | | 0.01 | (-0.35, 0.36) | | -0.20 | (-0.55, 0.16) | | -0.14 | (-0.48, 0.21) | | -0.08 | (-0.41, 0.25) | | -0.07 | (-0.42, 0.29) | |
| 41–50 | | 0.02 | (-0.59, 0.63) | | 0.17 | (-0.48, 0.81) | | -0.14 | (-0.78, 0.50) | | 0.34 | (-0.29, 0.97) | | 0.26 | (-0.35, 0.86) | | -0.12 | (-0.76, 0.52) | |
| 51–60 | | 1.28 | (0.47, 2.10) | ** | 1.32 | (0.46, 2.18) | ** | 0.65 | (-0.21, 1.50) | | 0.31 | (-0.53, 1.16) | | 0.89 | (0.08, 1.70) | * | 0.28 | (-0.58, 1.14) | |
| 61–70 | | 1.33 | (0.04, 2.61) | * | 0.82 | (-0.54, 2.18) | | 0.34 | (-1.01, 1.69) | | -0.25 | (-1.59, 1.08) | | -0.01 | (-1.28, 1.26) | | -0.13 | (-1.48, 1.23) | |
| 71–80 | | - | | | - | | | - | | | - | | | - | | | - | | |
| 81- | | -0.36 | (-1.65, 0.93) | | -0.55 | (-1.92, 0.82) | | 0.32 | (-1.04, 1.68) | | -0.66 | (-2.01, 0.68) | | -0.81 | (-2.09, 0.47) | | 0.31 | (-1.05, 1.67) | |

+ p<0.1

*p<0.05, ** p<0.01, ***p<0.001, Higher score indicates unfavorable stress-response. All betas were adjusted by age, type of job, job class, employment status, type of schedule, company size, company industry, job control, supervisors support and coworker's support. Lack of vigor was derived by reversing the score of vigor for harmonization with other stress-response scales, which higher score indicates unfavorable stress-response

workers. Due to the small number of working women in higher job-positions, the interaction by job-class was not clear. It is interesting that the highest beta coefficient in point was observed for "fatigue" among women in administrative positions (beta = 1.16 [0.68. 1.63]).

This study aimed to clarify the dose-response relationships between length of overtime and stress response using data of 59,021 Japanese workers. As a result, workers with longer overtime showed higher "irritability", "fatigue", "anxiety", "depression" and "somatic responses" than those with shorter overtime, adjusted for possible confounders. In addition, linear dose-response curves were observed between length of overtime and these stress responses. Whereas, only "lack of vigor" was not consistently associated with overtime. Rather, male workers with 61–80 hours of monthly overtime were more likely to feel vigorous than workers with shorter overtime.

As expected, stress response levels were relatively worse among workers with longer overtime than those with shorter hours. Several longitudinal studies have consistently showed that long working hours could elevate the risk of mental health issues, such as depression [16,18,32,33] or anxiety [16]. Especially, a recent systematic review shows that long working hours are related to higher risk of depression in Asian countries, compared to European countries [34]. Indeed working time regulation differed between European and Asian countries [35]. However, the present study showed results compatible with previous studies and reinforced the evidence that longer working hours could have harmful effects on workers' mental health.

The present study showed linear dose-response relations between length of overtime and stress responses. This linear, no-threshold association implies that even small reduction of overtime could be beneficial for workers to reduce their stress responses, which have been indicated as major risk factors for sickness absence among workers [36]. Therefore, it is important for health care managers and employers to shorten workers' overtime to prevent sick leave by alleviating psychological responses such as fatigue.

Here we briefly state possible mechanisms as to why severe stress responses were observed among workers with longer overtime. First, workers with long working hours need a sufficient amount of time for recovery from work, and these workers may not have enough time for

**Table 4. Association between overtime work hours and stress responses: Results of multiple linear regression stratified by gender and job-class.**

| Overtime work hours/month | n | Lack of Vigor | | | Irritability | | | Fatigue | | | Anxiety | | | Depression | | | Somatic responses | | |
|---|---|---|---|---|---|---|---|---|---|---|---|---|---|---|---|---|---|---|---|
| | | beta | 95% CI | p | beta | 95% CI | p | beta | 95% CI | p | beta | 95% CI | p | beta | 95% CI | p | beta | 95% CI | p |
| **Men** | | | | | | | | | | | | | | | | | | | |
| **Workers in managerial or directorial position** | | | | | | | | | | | | | | | | | | | |
| ≤20 | 2,117 | Ref | | | Ref | | | Ref | | | Ref | | | Ref | | | Ref | | |
| 21–30 | 1,602 | 0.07 | (0.01, 0.12) | * | 0.11 | (0.06, 0.17) | *** | 0.15 | (0.10, 0.21) | *** | 0.14 | (0.08, 0.19) | *** | 0.08 | (0.02, 0.13) | ** | 0.04 | (-0.02, 0.09) | |
| 31–40 | 1,639 | 0.02 | (-0.04, 0.07) | | 0.12 | (0.06, 0.18) | *** | 0.27 | (0.21, 0.32) | *** | 0.24 | (0.18, 0.29) | *** | 0.13 | (0.07, 0.18) | *** | 0.07 | (0.01, 0.12) | * |
| 41–50 | 1,475 | -0.03 | (-0.09, 0.03) | | 0.17 | (0.11, 0.23) | *** | 0.31 | (0.25, 0.37) | *** | 0.28 | (0.22, 0.34) | *** | 0.11 | (0.05, 0.17) | *** | 0.06 | (-0.01, 0.12) | + |
| 51–60 | 1,339 | -0.03 | (-0.09, 0.04) | | 0.19 | (0.12, 0.25) | *** | 0.39 | (0.33, 0.45) | *** | 0.32 | (0.25, 0.38) | *** | 0.12 | (0.06, 0.18) | *** | 0.08 | (0.02, 0.15) | * |
| 61–70 | 832 | -0.12 | (-0.19, -0.05) | ** | 0.16 | (0.09, 0.23) | *** | 0.43 | (0.35, 0.50) | *** | 0.33 | (0.26, 0.40) | *** | 0.09 | (0.02, 0.16) | ** | 0.02 | (-0.06, 0.09) | |
| 71–80 | 915 | -0.08 | (-0.15, -0.01) | * | 0.31 | (0.24, 0.38) | *** | 0.56 | (0.49, 0.63) | *** | 0.47 | (0.40, 0.54) | *** | 0.18 | (0.11, 0.25) | *** | 0.13 | (0.06, 0.21) | *** |
| 81- | 430 | -0.02 | (-0.11, 0.07) | | 0.33 | (0.24, 0.42) | *** | 0.72 | (0.63, 0.81) | *** | 0.47 | (0.38, 0.56) | *** | 0.23 | (0.14, 0.32) | *** | 0.11 | (0.01, 0.20) | * |
| p for linear trend | | <0.001 (negative) | | | <0.001 (positive) | | | <0.001 (positive) | | | <0.001 (positive) | | | <0.001 (positive) | | | <0.001 (positive) | | |
| **Workers in regular position** | | | | | | | | | | | | | | | | | | | |
| ≤20 | 13,634 | Ref | | | Ref | | | Ref | | | Ref | | | Ref | | | Ref | | |
| 21–30 | 5,658 | 0.04 | (0.01, 0.07) | * | 0.12 | (0.09, 0.15) | *** | 0.17 | (0.14, 0.20) | *** | 0.10 | (0.07, 0.13) | *** | 0.04 | (0.01, 0.07) | ** | 0.02 | (-0.01, 0.05) | |
| 31–40 | 4,173 | 0.01 | (-0.02, 0.04) | | 0.16 | (0.13, 0.19) | *** | 0.27 | (0.24, 0.30) | *** | 0.13 | (0.10, 0.16) | *** | 0.04 | (0.01, 0.08) | * | 0.07 | (0.03, 0.10) | *** |
| 41–50 | 2,622 | 0.02 | (-0.02, 0.06) | | 0.16 | (0.12, 0.20) | *** | 0.41 | (0.37, 0.44) | *** | 0.20 | (0.16, 0.24) | *** | 0.14 | (0.10, 0.18) | *** | 0.17 | (0.13, 0.21) | *** |
| 51–60 | 2,155 | -0.04 | (-0.08, 0.00) | + | 0.18 | (0.14, 0.23) | *** | 0.44 | (0.40, 0.49) | *** | 0.22 | (0.18, 0.27) | *** | 0.12 | (0.08, 0.16) | *** | 0.16 | (0.11, 0.20) | *** |
| 61–70 | 982 | -0.13 | (-0.19, -0.07) | *** | 0.23 | (0.16, 0.29) | *** | 0.55 | (0.49, 0.61) | *** | 0.41 | (0.35, 0.47) | *** | 0.26 | (0.19, 0.32) | *** | 0.26 | (0.20, 0.32) | *** |
| 71–80 | 825 | -0.18 | (-0.24, -0.11) | *** | 0.25 | (0.18, 0.31) | *** | 0.61 | (0.54, 0.67) | *** | 0.42 | (0.36, 0.49) | *** | 0.31 | (0.24, 0.38) | *** | 0.30 | (0.23, 0.37) | *** |
| 81- | 366 | 0.01 | (-0.08, 0.10) | | 0.42 | (0.33, 0.52) | *** | 0.76 | (0.67, 0.86) | *** | 0.37 | (0.27, 0.47) | *** | 0.31 | (0.21, 0.41) | *** | 0.34 | (0.24, 0.44) | *** |
| p for linear trend | | <0.001 (negative) | | | <0.001 (positive) | | | <0.001 (positive) | | | <0.001 (positive) | | | <0.001 (positive) | | | <0.001 (positive) | | |
| **Interaction between overtime and age-category** | | | | | | | | | | | | | | | | | | | |
| **Regular (ref) versus Manager** | | | | | | | | | | | | | | | | | | | |
| 21–30 | | 0.03 | (-0.03, 0.09) | | 0.00 | (-0.07, 0.06) | | -0.01 | (-0.07, 0.06) | | 0.03 | (-0.03, 0.10) | | 0.02 | (-0.04, 0.09) | | 0.02 | (-0.05, 0.08) | |
| 31–40 | | 0.00 | (-0.06, 0.07) | | -0.02 | (-0.09, 0.04) | | 0.01 | (-0.06, 0.07) | | 0.10 | (0.04, 0.17) | ** | 0.07 | (0.00, 0.14) | * | 0.01 | (-0.06, 0.07) | |
| 41–50 | | -0.05 | (-0.12, 0.02) | | 0.02 | (-0.05, 0.09) | | -0.09 | (-0.16, -0.02) | * | 0.07 | (0.00, 0.14) | * | -0.06 | (-0.13, 0.02) | | -0.10 | (-0.17, -0.02) | ** |
| 51–60 | | 0.01 | (-0.06, 0.08) | | 0.04 | (-0.04, 0.11) | | -0.04 | (-0.11, 0.03) | | 0.08 | (0.00, 0.15) | * | -0.04 | (-0.11, 0.04) | | -0.04 | (-0.12, 0.03) | |
| 61–70 | | 0.00 | (-0.09, 0.09) | | -0.03 | (-0.12, 0.07) | | -0.11 | (-0.20, -0.02) | * | -0.10 | (-0.19, 0.00) | * | -0.19 | (-0.29, -0.10) | *** | -0.22 | (-0.31, -0.12) | *** |
| 71–80 | | 0.09 | (0.00, 0.18) | + | 0.10 | (0.01, 0.20) | * | -0.04 | (-0.13, 0.05) | | 0.02 | (-0.08, 0.11) | | -0.16 | (-0.26, -0.07) | ** | -0.13 | (-0.23, -0.03) | ** |
| 81- | | 0.01 | (-0.12, 0.14) | | -0.09 | (-0.22, 0.05) | | -0.04 | (-0.17, 0.10) | | 0.09 | (-0.04, 0.22) | | -0.10 | (-0.23, 0.03) | | -0.23 | (-0.37, -0.09) | |

*(Continued)*

**Table 4.** (Continued)

| Overtime work hours/month | n | Lack of Vigor | | | Irritability | | | Fatigue | | | Anxiety | | | Depression | | | Somatic responses | | |
|---|---|---|---|---|---|---|---|---|---|---|---|---|---|---|---|---|---|---|---|
| | | beta | 95% CI | p | beta | 95% CI | p | beta | 95% CI | p | beta | 95% CI | p | beta | 95% CI | p | beta | 95% CI | p |
| **Women** | | | | | | | | | | | | | | | | | | | |
| **Workers in managerial or directorial position** | | | | | | | | | | | | | | | | | | | |
| ≤20 | 151 | Ref | | | Ref | | | Ref | | | Ref | | | Ref | | | Ref | | |
| 21–30 | 124 | 0.08 | (-0.15, 0.31) | | 0.04 | (-0.18, 0.25) | | 0.04 | (-0.19, 0.27) | | -0.01 | (-0.23, 0.21) | | 0.02 | (-0.19, 0.23) | | 0.04 | (-0.18, 0.26) | |
| 31–40 | 125 | 0.08 | (-0.16, 0.31) | | 0.23 | (0.01, 0.45) | * | 0.43 | (0.20, 0.67) | *** | 0.48 | (0.26, 0.71) | *** | 0.17 | (-0.05, 0.38) | | 0.28 | (0.05, 0.50) | * |
| 41–50 | 81 | 0.23 | (-0.04, 0.49) | + | 0.09 | (-0.16, 0.34) | | 0.35 | (0.09, 0.62) | ** | 0.33 | (0.07, 0.58) | * | 0.03 | (-0.21, 0.27) | | 0.10 | (-0.16, 0.35) | |
| 51–60 | 64 | 0.01 | (-0.28, 0.31) | | 0.10 | (-0.17, 0.37) | | 0.39 | (0.09, 0.68) | ** | 0.28 | (0.00, 0.56) | + | 0.03 | (-0.23, 0.30) | | 0.14 | (-0.14, 0.43) | |
| 61–70 | 33 | 0.43 | (0.06, 0.80) | * | 0.46 | (0.11, 0.80) | *** | 0.86 | (0.49, 1.23) | *** | 0.85 | (0.50, 1.21) | *** | 0.39 | (0.06, 0.72) | ** | 0.28 | (-0.08, 0.64) | |
| 71–80 | 18 | 0.10 | (-0.38, 0.57) | | 0.23 | (-0.22, 0.67) | | 1.16 | (0.68, 1.63) | *** | 0.70 | (0.25, 1.15) | ** | 0.77 | (0.34, 1.19) | *** | 0.66 | (0.20, 1.12) | ** |
| 81- | 20 | -0.04 | (-0.49, 0.40) | | 0.22 | (-0.20, 0.63) | | 0.52 | (0.07, 0.97) | * | 0.43 | (0.00, 0.86) | * | 0.09 | (-0.31, 0.50) | | 0.20 | (-0.24, 0.63) | |
| p for linear trend | | n.s. | | | <0.001 (positive) | | | <0.001 (positive) | | | <0.001 (positive) | | | <0.05 (positive) | | | <0.05 (positive) | | |
| **Workers in regular position** | | | | | | | | | | | | | | | | | | | |
| ≤20 | 13,378 | Ref | | | Ref | | | Ref | | | Ref | | | Ref | | | Ref | | |
| 21–30 | 2,177 | 0.07 | (0.03, 0.12) | ** | 0.15 | (0.11, 0.19) | *** | 0.27 | (0.23, 0.32) | *** | 0.20 | (0.15, 0.24) | *** | 0.14 | (0.09, 0.18) | *** | 0.12 | (0.08, 0.17) | *** |
| 31–40 | 1,094 | 0.16 | (0.10, 0.22) | *** | 0.22 | (0.16, 0.28) | *** | 0.47 | (0.41, 0.53) | *** | 0.35 | (0.29, 0.40) | *** | 0.20 | (0.14, 0.26) | *** | 0.19 | (0.13, 0.25) | *** |
| 41–50 | 486 | 0.13 | (0.04, 0.21) | ** | 0.22 | (0.13, 0.30) | *** | 0.54 | (0.45, 0.63) | *** | 0.34 | (0.25, 0.42) | *** | 0.24 | (0.16, 0.33) | *** | 0.20 | (0.12, 0.29) | *** |
| 51–60 | 262 | -0.05 | (-0.17, 0.06) | | 0.22 | (0.10, 0.34) | *** | 0.63 | (0.51, 0.75) | *** | 0.55 | (0.43, 0.66) | *** | 0.39 | (0.28, 0.51) | *** | 0.29 | (0.17, 0.41) | *** |
| 61–70 | 94 | 0.03 | (-0.16, 0.22) | | 0.16 | (-0.04, 0.36) | | 0.83 | (0.63, 1.02) | *** | 0.66 | (0.46, 0.85) | *** | 0.57 | (0.39, 0.76) | *** | 0.48 | (0.29, 0.68) | *** |
| 71–80 | 79 | -0.19 | (-0.39, 0.02) | + | 0.40 | (0.18, 0.61) | *** | 0.70 | (0.48, 0.91) | *** | 0.70 | (0.48, 0.91) | *** | 0.55 | (0.35, 0.75) | *** | 0.39 | (0.18, 0.61) | *** |
| 81- | 71 | 0.05 | (-0.17, 0.26) | | 0.33 | (0.10, 0.55) | ** | 0.64 | (0.42, 0.87) | *** | 0.53 | (0.31, 0.75) | *** | 0.38 | (0.17, 0.60) | *** | 0.13 | (-0.10, 0.35) | |
| p for linear trend | | <0.001 (positive) | | | <0.001 (positive) | | | <0.001 (positive) | | | <0.001 (positive) | | | <0.001 (positive) | | | <0.001 (positive) | | |
| **Interaction between overtime and age-category** | | | | | | | | | | | | | | | | | | | |
| **Regular (ref) versus Manager** | | | | | | | | | | | | | | | | | | | |
| 21–30 | | -0.03 | (-0.25, 0.19) | | -0.11 | (-0.34, 0.13) | | -0.30 | (-0.53, -0.07) | * | -0.23 | (-0.46, 0.00) | + | -0.15 | (-0.37, 0.07) | | -0.09 | (-0.33, 0.14) | |
| 31–40 | | -0.14 | (-0.37, 0.09) | | 0.02 | (-0.22, 0.26) | | -0.13 | (-0.36, 0.11) | | 0.12 | (-0.12, 0.35) | | -0.06 | (-0.28, 0.16) | | 0.05 | (-0.19, 0.28) | |
| 41–50 | | 0.06 | (-0.20, 0.33) | | -0.14 | (-0.41, 0.14) | | -0.25 | (-0.52, 0.03) | + | -0.03 | (-0.30, 0.24) | | -0.24 | (-0.49, 0.02) | + | -0.15 | (-0.43, 0.12) | |
| 51–60 | | 0.01 | (-0.28, 0.31) | | -0.07 | (-0.37, 0.24) | | -0.31 | (-0.61, 0.00) | * | -0.28 | (-0.58, 0.02) | + | -0.40 | (-0.68, -0.11) | ** | -0.19 | (-0.49, 0.12) | |
| 61–70 | | 0.29 | (-0.11, 0.68) | | 0.29 | (-0.12, 0.71) | | -0.14 | (-0.55, 0.28) | | 0.14 | (-0.27, 0.55) | | -0.23 | (-0.62, 0.16) | | -0.29 | (-0.71, 0.12) | |
| 71–80 | | 0.19 | (-0.31, 0.68) | | -0.09 | (-0.61, 0.43) | | 0.35 | (-0.17, 0.86) | | 0.02 | (-0.49, 0.53) | | 0.19 | (-0.29, 0.68) | | 0.26 | (-0.26, 0.78) | |

(*Continued*)

**Table 4.** (Continued)

| Overtime work hours/month | n | Lack of Vigor | | | Irritability | | | Fatigue | | | Anxiety | | | Depression | | | Somatic responses | | |
|---|---|---|---|---|---|---|---|---|---|---|---|---|---|---|---|---|---|---|---|
| | | beta | 95% CI | p | beta | 95% CI | p | beta | 95% CI | p | beta | 95% CI | p | beta | 95% CI | p | beta | 95% CI | p |
| 81- | | -0.20 | (-0.68, 0.28) | | -0.09 | (-0.60, 0.42) | | -0.19 | (-0.70, 0.31) | | -0.09 | (-0.58, 0.41) | | -0.30 | (-0.77, 0.17) | | 0.03 | (-0.48, 0.53) | |

+ p<0.1

*p<0.05, ** p<0.01, ***p<0.001, Higher score indicates unfavorable stress-response. All betas were adjusted by age, type of job, employment status, type of schedule, company size, company industry, job control, supervisors support and coworker's support. Lack of vigor was derived by reversing the score of vigor for harmonization with other stress-response scales, which higher score indicates unfavorable stress-response

relaxing or refreshing themselves after finishing work. This may be one possible reason why older workers showed higher stress responses than workers in middle age among with 21–50 hours/month of overtime. Since older adults may need more recovery time, they may be more vulnerable for overtime working. Furthermore, long working hours also lead to short sleeping hours and deteriorated sleep quality, such as difficulty in falling asleep or waking without feeling refreshed [8,37]. Avoiding overly long working hours is important to prevent workers' mental health problems by allowing time for good sleeping conditions and refreshment.

Unexpectedly, "lack of vigor" was shown to have a distinct relationship to overtime, which differed from other types of stress responses. Particularly, men with 61–80 hours of overtime showed relatively higher vigor level, together with severe fatigue or anxiety levels at the same time. The detailed mechanisms for this ambivalence were unknown. Referring to the classical stress model introduced by Henry [38], in early to middle stages of the stress response process, stress stimuli may raise vigorous feeling through hormonal mechanisms. A similar phenomenon is also known as "runner's high" in exercise research [39]. When participants received excessive stress through endurance exercise (i.e. triathlon), almost all participants felt fatigue. Nevertheless, some participants kept their vigorous state and showed increased levels of adrenocorticotropic hormone and beta-endorphin [40,41]. One study of 217 Korean male workers reported a marginally-positive correlation between working hours and adrenaline in urine [42]. Furthermore, association of stress-induced cortisol elevation and low poststress negative affect ratings was also reported [43]. Taken together, it may be speculated that workers with 61–80 hours/month of overtime, who showed high vigor and high fatigue at the same time, may experience a similar state to "runners' high". Since the present study did not include any biological data, this state may need to be verified in future studies which use detailed clinical data. In addition, the increase of vigor among male workers with 61–80 hours/month of overtime may reflect their overcommitment to work. Workers with overcommitment often exaggerate their efforts beyond appropriate levels, showing temporal and energetic characteristics of behavior [44]. However, it is reported that such efforts weaken their potential recovery from job demands and increase their frustration when the expected rewards are not fulfilled, which eventually results in poor health such as cardiovascular disease [45] or depression [46]. This unfavorable condition may occur among male workers with 81 or more hours/month of overtime, who showed rather low levels of vigor. Consequently, the total relationship between overtime work hours and "lack of vigor" created an inverse U-shape curve, which is the same as the stress response curve in Henry's theory, where the last phase is called "overload state" [38]. Due to the cross-sectional study design, the present study could not consider the longitudinal effect of working hours and changes in stress responses. However, it may be possible that a

distinguishing increase in vigor may cause a loss of self-control of workload, resulting in excessively long work. Such workers would be in a physically and mentally exhausted state, that is, they would be at a high risk for *Karoshi* or *Karojisatsu*.

A positive linear association between length of overtime and vigor was found among men, whereas a negative linear association was found among women. Specific reasons for this gender-gap are unknown. However, self-selection into job with long working hours among men could be speculated. As discussed above, workers with high vigor may aspire to work longer hours. However, that may be difficult for women due to household responsibilities or other familial roles. Hence, in comparison to vigorous men, vigorous women may have relatively shorter length of working hours, which may be one possible cause of this gender-gap. The difference in work-family spillover between working men and women could be suggested as another possible explanator factor. According to the Japanese Survey on Time use and Leisure activities in 2016, among couples with preschool children, the average time for child rearing or household chores were 84 min/week and 370 min/week for men and women, respectively [47]. It is reported that work family negative spillover is more explicit in women, compared with men, and is related with psychological distress [48]. Speculatively, women with long working hours may tend to suffer more from negative work-family spillover, compared to men. Finally, denial and dissimulation may be an increasing problem with increasing number of working hours. This may be a frequent problem in men who want to stand out as invincible. In fact, denial has been pointed out as a possible risk factor for sudden cardiac death [49] although this is hard to examine in epidemiological studies and cannot be studied at all in cross-sectional examinations.

Vigor among workers have attracted research attention in positive occupational health psychology for instance in studies of work engagement [50,51]. Work engagement has been defined as "*a positive, fulfilling, work-related state of mind that is characterized by vigor, dedication and absorption*" [52]. Vigor in work-engagement assessments has included concepts of "motivation" or "psychological resilience", whereas the items for assessment of vigor in BJSQ have been based on POMS. According to Shirom, the conceptualization of vigor in both work-engagement and BJSQ is not exactly the same [53]. In our study "lack of vigor" was measured from the POMS. This negative concept is related on a deeper theoretical level to fatigue. In fact, moderate correlations between "lack of vigor" and "fatigue" were observed (men: r = -0.328, women: r = -0.358). However, the distinction between these interrelated concepts remains unclear theoretically. Exploring the association between overtime work hours and vigor in work engagement is an important research question in future studies.

The possible reason why high vigor were observed among men with longer working hours is a high-level of work-engagement. In addition to dedication and absorption, vigor was conceptually one component of work-engagement [50,51]. In addition, another study showed that excess work engagement was observed in the population with long working hours [54]. The data in the present study did not include any data regarding work-engagement. However, future studies including work engagement may be needed to clarify the distinct association between vigor and overtime work hours among men.

Currently, the Labor Standards Law in Japan technically permits "no limitation of overwork" under article 36 of the law, which allows managers to enforce as much overtime as they think is needed to meet the production target [11]. In response to public criticism of unlimited overtime, the Japanese government recently attempted to amend the law to introduce an overtime limit of both 100 hours/month and 720 hours/year [13]. In the present study, workers with 61–80 hours/month of overtime may be regarded as "full of energy", but they also have severe fatigue levels. Thus, this regulation criteria may cause a misimpression that they are not a risky population, and consequently, they may engage in additional overwork. Therefore, not

only self-management of working hours, but also regulation by law is important to prevent overtime and its related mental illnesses among these workers. Taking this into account, the regulation of 100 hours/month may not be sufficient.

The specific favorable characteristics of the present study are a relatively large sample-size and the fact that participants were selected from different industries and company-sizes. However, some limitations of our study should also be considered. First, a cross-sectional study design limits arguing causality between length of overtime and stress response levels. It is possible that workers with high vigor may have strong motivation to work because of energetic characteristics, that lead to work longer. However, the cross-sectional study design cannot identify the direction of the selection process. To clarify how stress response level and length of overtime are associated or influenced by each other, longitudinal studies are needed. Second, our analysis did not consider individual past/current disease history. It is possible that some workers need to restrict their working hours due to receiving medication at the point of survey. If their stress response levels are higher than other healthy workers even though their overtime work hours is relatively short, the result of the present study may be underestimated. In addition, we did not consider personal characteristics or familial history of mental health. For example, the results of this study may suffer from confounding due to type A behavioral patterns, where workers with such behavioral patterns tend to work long hours with high tension [55]. The results of this study may suffer from confounding due to such unobserved factors. Third, a relatively substantial number of subjects (n = 8,560) were excluded due to missing data. We cannot deny the possibility that workers with higher stress-response are more likely to have missing data or to avoid answering some questions on the BJSQ. If workers with longer overtime and higher stress responses coincide to those with missing data, the results would suffer from underestimation. Fourth, this study did not measure total working hours. Even though this study excluded shift-workers and part-timers, it is possible that it included workers whose regular working hours are relatively short. Their total working hours may not be long even if they reported very long overtime. In addition, overtime work hours were measured only once. Since working hours are not necessarily stable over a long period, our results may be affected by potential misclassification. Whereas, a recent longitudinal study among 18,172 Japanese workers showed that 85% of subjects reported stable overtime work hours for three consecutive years [56], thus misclassification of working hours may not be problematic. Furthermore, overtime work hours were assessed by self-report. Although a previous study showed that measuring overtime through self-report had high validity and that reproducibility was confirmed among Japanese workers [57], self-reported overtime work hours may be less accurate than objectively-measured working hours. In addition, this study did not consider work pace of overtime. Even among workers with the same length of overtime working, a higher stress response may be observed among those who are under severe time-pressure or experience a higher working pace [58]. Fifth, the present study could not perform a nested data analysis due to lack of detailed workplace information. Individual stress responses could be affected not only on the individual level, but also on the group and workplace level [59], thus future studies using multilevel analysis with detailed workplace data would be preferable [60]. Finally, the data in the present study may suffer from anticipation bias. Since participants received the Stress Check Program in non-anonymous manner, they may report their stress levels with anticipated responses, and it may not sufficiently reflect the real situation.

## Conclusions

Length of overtime shows linear associations with various psychosomatic stress responses. However, "lack of vigor" was not consistently associated with overtime. Male workers with 61–

80 hours of monthly overtime were more likely to feel vigorous than workers with shorter overtime. However, potential longterm effects of such extreme overtime should not be underestimated and must be paid attention to.

## Supporting information

**S1 Table.** A Specific items with internal consistency of the scales of Brief Job Stress Questionnaire used in the study. B Correlation coefficient between stress response scales in BJSQ. (DOCX)

**S2 Table. Association between overtime work hours and stress responses: Multiple linear regression results with multiple imputation.**
(DOCX)

## Acknowledgments

We thank Dr. Keisuke Fukui at Osaka Medical College for helpful comments to statistical analysis.

## Author Contributions

**Conceptualization:** Hiroyuki Kikuchi, Yumiko Ohya, Teruichi Shimomitsu.

**Data curation:** Hiroyuki Kikuchi, Yumiko Ohya, Yutaka Nakanishi, Teruichi Shimomitsu.

**Formal analysis:** Hiroyuki Kikuchi.

**Investigation:** Yuko Odagiri, Yumiko Ohya.

**Resources:** Yutaka Nakanishi, Teruichi Shimomitsu.

**Software:** Yutaka Nakanishi, Teruichi Shimomitsu.

**Supervision:** Yuko Odagiri, Teruichi Shimomitsu, Töres Theorell, Shigeru Inoue.

**Writing – original draft:** Hiroyuki Kikuchi.

**Writing – review & editing:** Yuko Odagiri, Yumiko Ohya, Teruichi Shimomitsu, Töres Theorell, Shigeru Inoue.

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
