## [Decision Letter · Decision Letter 0]

30 Sep 2019

PONE-D-19-24855

Distinct features in association of overtime work hours with various stress responses: results from 59,021 Japanese workers’ cross-sectional study

PLOS ONE

Dear Dr. Odagiri,

Thank you for submitting your manuscript to PLOS ONE. After careful consideration, we feel that it has merit but does not fully meet PLOS ONE’s publication criteria as it currently stands. Therefore, we invite you to submit a revised version of the manuscript that addresses the points raised during the review process.

The two reviewers addressed a number of major and minor concerns about your manuscript. Please revise your manuscript carefully.

We would appreciate receiving your revised manuscript by Nov 14 2019 11:59PM. To enhance the reproducibility of your results, we recommend that if applicable you deposit your laboratory protocols in protocols.io, where a protocol can be assigned its own identifier (DOI) such that it can be cited independently in the future. For instructions see: http://journals.plos.org/plosone/s/submission-guidelines#loc-laboratory-protocols

We look forward to receiving your revised manuscript.

Kind regards,

Kenji Hashimoto, PhD

Academic Editor

PLOS ONE

Journal Requirements:

2. Please include additional information regarding the survey or questionnaire used in the study and ensure that you have provided sufficient details that others could replicate the analyses. For instance, if you developed a questionnaire as part of this study and it is not under a copyright more restrictive than CC-BY, please include a copy, in both the original language and English, as Supporting Information. Furthermore, please refer to any post-hoc corrections for multiple comparisons that were made following your statistical analysis. If these were not performed please justify why.

Additional Editor Comments (if provided):

Reviewers' comments:

Reviewer's Responses to Questions

**Comments to the Author**

1. Is the manuscript technically sound, and do the data support the conclusions?

Reviewer #1: Partly

Reviewer #2: Partly

2. Has the statistical analysis been performed appropriately and rigorously? 

Reviewer #1: I Don't Know

Reviewer #2: Yes

3. Have the authors made all data underlying the findings in their manuscript fully available?

Reviewer #1: No

Reviewer #2: No

4. Is the manuscript presented in an intelligible fashion and written in standard English?

Reviewer #1: Yes

Reviewer #2: Yes

5. Review Comments to the Author

Reviewer #1: I have read the paper entitled „Distinct features in association of overtime work hours with various stress responses: results from 59,021 Japanese workers’ cross-sectional study” submitted to PLOSone with great interest. The study describes patterns of association between hours worked overtime and several indicators of strain or impaired (mental) health. Below I raise a couple of questions, that need to be clarified. I hope my comments are helpful in improving the paper.

1. Theory

1.1. As far as I have understood the PLOSone does not emphasizes *theoretical* contributions to the literature. However, there are a couple of meta-analyses and large survey studies about overtime. Hence, stating clearly how this study adds to the literature would be beneficial. You argue that your sample is stratified and the others were not, but why is that important?

1.2. On a related note, you seem to provide a more fine-grained perspective on hours of overtime. Are there any inconsistencies that would suggest taking a closer look? In other words, besides the technical aspect: How does your approach advance knowledge? Does it provide a more accurate picture? Does it help to gain a more integrated picture (comparing many different types of strain in one study)? Does it facilitate comparison between strain indicators? Does it add data on Japanese employees? I recommend stating more clearly how this study enriches existing research. I am quite sure it does.

1.3. The choice of strain measures is not justified at all. They seem to be taken for granted. However, it is not clear why exactly those aspects were chosen. I understand that this has been discussed in the other study using these data. For readers who do not want to take the effort of looking up details elsewhere, please add a short description of the strain indicators. What characterizes fatigue in the sense you mean? How does it differ from lack of vigor both conceptually and empirically? Is “lack of vigor” the opposite of vigor? I found the expression quite complicated, as it includes a negation. Please also note that there is a long list of papers on the question whether vigor is the opposite of exhaustion (work engagement vs. burnout) (Demerouti, Mostert, & Bakker, 2010; González-Romá, Schaufeli, Bakker, & Lloret, 2006).

1.4. Please also add a more systematic review of how overtime is related to fatigue, lack of vigor, depression etc. respectively. These details would help judge your results in the light of prior research. Is irritability the same thing as irritation (Mohr, Müller, Rigotti, Aycan, & Tschan, 2006)? I think, it is important to locate all variables in the nomological network and to be clear what these concepts actually mean.

2. Methods

2.1. Please report sample items for each scale. Please also describe the scope of each scale.

2.2. You excluded participants from small organizations and participants who did not receive the program? What does receiving the program mean? It is not trivial to me why these groups were excluded?

2.3. You exclude a portion of 15 per cent because of missing data. Which data were missing? For instance, if only single items or single scales were missing by these persons, wouldn’t inclusion of all available data be better? At a minimum, I would make a case that exclusion of these cases does not affect your results and conclusions. For instance, you could compare the focal (conservative) vs. the larger sample applying a more liberal approach of exclusion.

2.4. Do you have information on nesting of data (which persons belonged to which organization)? If yes, please examine the degree of nesting of data (Luke, 2004). If nesting is an issue you should take nesting into account by virtue of multilevel modeling. IF you lack information, please discuss ignorance of nesting of the data as a limitation of your study.

2.5. I commend the comprehensive description of the sample.

3. Results

3.1. Please provide a correlation table including all focal variables. Include means, standard deviations, and a measure of reliability of multi-item scales, such as Cronbachs Alpha or Omega. This would facilitate inclusion of your study in meta-analyses. This would also help understand the counter-intuitive results for “lack of vigor”. I would expect fatigue and vigor to be highly negatively correlated. However, in the work engagement literature there are also aspects like being very resilient while working intensively (Mills, Culbertson, & Fullagar, 2012).

3.2. What does p for trend mean?

3.3. It was not totally clear to me what you did with the ANOVA. What were the factors? Were the classes of overtime hours one factor? Would an omnibus test really be the best approach? Given your specific expectations, a contrast analysis might be viable, too. As far as I understood Table 2, you compared all classes to the reference group of no or few hours overtime. A contrast analysis would help being more specific an allow comparisons between all groups.

3.4. Given your high number of significance tests, probably a Bonferroni-correction or something would be warranted. Given the probably very high test power, have you considered reporting effect sizes? I mean with a sample of 50,000+ almost any small effect likely will be significant.

3.5. I think, you mention regression analysis in the methods section. What did the regression model look like? What did you analyze? Where are the results reported?

4. Discussion

4.1. I appreciate the comprehensive discussion of results and the acknowledgment of limitations.

Demerouti, E., Mostert, K., & Bakker, A. B. (2010). Burnout and work engagement: A thorough investigation of the independency of both constructs. Journal of Occupational Health Psychology, 15(3), 209–222. https://doi.org/10.1037/a0019408

González-Romá, V., Schaufeli, W. B., Bakker, A. B., & Lloret, S. (2006). Burnout and work engagement: Independent factors or opposite poles? Journal of Vocational Behavior, 68(1), 165–174. https://doi.org/10.1016/j.jvb.2005.01.003

Luke, D. A. (2004). Multilevel modeling. Thousand Oaks: Sage.

Mills, M. J., Culbertson, S. S., & Fullagar, C. J. (2012). Conceptualizing and measuring engagement: An analysis of the Utrecht Work Engagement Scale. Journal of Happiness Studies, 13(3), 519–545. https://doi.org/10.1007/s10902-011-9277-3

Mohr, G., Müller, A., Rigotti, T., Aycan, Z., & Tschan, F. (2006). The assessment of psychological strain in work contexts: Concerning the structural equivalency of nine language adaptations of the irritation scale. European Journal of Psychological Assessment, 22(3), 198–206. https://doi.org/10.1027/1015-5759.22.3.198

Reviewer #2: Manuscript ID: PONE-D-19-24855: Distinct features in association of overtime work hours with various stress responses: results from 59,021 Japanese workers’ cross-sectional study

General comments:

Important issue and large sample size make the attractive setting of this study. However, there are minor revsions to be established.

The authors present a retrospective evaluation of data from the national stress check program. In order to give feedback to participant’s data were collected in a non-anonymized manner. This needs to be specified more clearly in methods section and needs to be discussed in limitations. Subjective assessment in self-reporting measures might reveal anticipated responses and not the real situation. Most likely anticipation-bias might have influenced the responses of workers in a relationship of professional dependency.

The BJSQ in its original form comprises 17 items on job stress, 29 items on psychological and physical stress reactions and 11 items on social support at work. (See: Tsutsumi A, Inoue A, Eguchi H. J Occup Health. 2017; 59(4): 356–360.)

Please specify whether you focused on the psychological and physical stress reactions with 29 items.

Specific comments:

Page 1, title

Specify in the title that this is a retrospective cross-sectional study.

Suggested wording: “Association of overtime work hours with various stress responses in 59,021 Japanese workers: retrospective cross-sectional study”

Page 3, abstract

Lines 35, 42…: Suggested phrasing: “overtime work” e.g. length of overtime work

Page 3, abstract

Line 37: Suggested phrasing: “self-reporting”

Page 5, introduction

Line 58: Delete: “too much”

Page 6, study design

Line 81: To my understanding, this is a retrospective data evaluation of non-anonymized data acquired by the Stress Check Program.

Please specify primary and secondary study objectives

Page 7, participants and data collection

Line 95: Consider phrasing: “eligible participants” …

Line 96: December 2015 to January 2016, …

Provide further information on how eligible participants were contacted, e.g. did the health service company support you with epidemiologic data and mail addresses?

Were non-respondents repeatedly contacted?

Page 8, stress responses

Line 119: Please specify whether you focused on the psychological and physical stress reactions with 29 items.

It would be helpful to the reader to know the original wording of the 29 items. Please enclose the questionnaire as appendix or as supplementary information to this manuscript.

Page 9, covariates

Line 131: Does history of diseases mean past medical history? This would mean a detailed register of previous diseases and provide the base for correlations between general health and number of overtime hours per month.

Page 9, statistical procedure

Line 140: The authors used multiple linear regression fitting a linear equation to independent variables and associated dependent variables. Multiple imputation was used for missing data management.

As there is known confounding by gender, qualification, leading position and age group I suggest to evaluate subgroups male/female in leading/non-leading position and age groups, in particular <30, 31-59, and >60 years (the range was 17 to 89 years).

Ordinal variables could be analyzed with the Mann-Whitney U test (n=2) and the Kruskal-Wallis test (n>2) in subgroup analysis.

Page 11, results and discussions

Line 164: Suggested wording: “did not receive” …

Line 170: Please distinguish between response rate and inclusion rate.

Out of 95,004 eligible workers, 88.988 were contacted of whom 83,470 responded (response rate: 87.9%). After exclusion of 24,449 workers, a total of 59,021 participants were included (inclusion rate: 62.1%)

Page 12, Table 1

It should be mentioned in results that the core group comprised middle-aged, male clerk with regular employment status and inflexible type of schedule. Consider that majority of workers reported overtime work of less than 20 hours a month.

Page 13, Table 1

Most commonly observed was manufacture with company size exceeding 3,000 employees.

Page 14

Results of questionnaires are based on subjective assessments. Better write “somebody reported” instead of “somebody was”.

A subgroup evaluation of age groups, in particular <30, 31-59, and >60 years might reveal striking differences in perceived fatigue and lack of vigor.

Line 188: suggested wording: “perceived” in perceived lack of vigor” …

Line 193: suggested wording: “reported” in reported to be more vigorous at work …

Line 197: suggested wording: “reported” in reported significantly higher …

Page 15

Line 204: When discussing differences between man and women perform a subgroup evaluation focusing on gender distribution in leading/non-leading positions.

Line 206: When taking about selection-bias arising from different health conditions consider that you lack profound variables on physical and mental health.

Page 16/17, table 2

In Characteristics: suggested wording: Overtime work hours/month

Page 18

Line 223: Take into account that overtime work not necessarily implies being under time pressure and working at high speed.

Line 230: Health perceptions differ according to age and cultural background and cannot be considered independently of a personal sense of vigor.

Consider that European Working Time Directives limit daily working time to 11 hours and weekly working time to 48 hours. (http://ec.europa.eu/social/main.jsp?catId=706&langId=en&intPageId=205).

Page 19

Line 241: Consider that the lifespan perspective of work design and ageing at work includes modification of job demands in relation to age

Page 21

Line: 280: Consider that effects of work-to-family spillover and from job demands that interfere with the family domain involve men and women.

6. PLOS authors have the option to publish the peer review history of their article (what does this mean?). If published, this will include your full peer review and any attached files.

Reviewer #1: Yes: Oliver Weigelt

Reviewer #2: No

---

## [Author Response · Author response to Decision Letter 0]

9 Dec 2019

Dear Prof. Heber and Dr.Hashimoto

Thank you for giving us the opportunity to revise and resubmit this manuscript. 

We would like to thank the Reviewers for their time and their valuable comments, which help us to improve the quality of our manuscript greatly. Below is our specific response to each comment. 

Please note that sentences in italics are the original text; italics with underline denote the revised or added text. Since we have revised the title based on the suggestion from the reviewer #2'. Now the new title is "Association of overtime work hours with various stress responses in 59,021 Japanese workers: retrospective cross-sectional study".

Thank you again for considering our manuscript.

Yours sincerely,

Yuko Odagiri, M.D., Ph.D.

Department of Preventive Medicine and Public Health, Tokyo Medical University, Japan 

Mailing address: 6-1-1, Shinjuku, Shinjuku-ku, Tokyo, 160-8402, Japan

Telephone: +81-3-5269-9785, Fax:+ 81-3-3353-0162

E-mail: odagiri@tokyo-med.ac.jp

---

## [Decision Letter · Decision Letter 1]

31 Dec 2019

PONE-D-19-24855R1

Association of overtime work hours with various stress responses in 59,021 Japanese workers: retrospective cross-sectional study

PLOS ONE

Dear Dr. Odagiri,

Thank you for submitting your manuscript to PLOS ONE. After careful consideration, we feel that it has merit but does not fully meet PLOS ONE’s publication criteria as it currently stands. Therefore, we invite you to submit a revised version of the manuscript that addresses the points raised during the review process.

The reviewer #1 addressed several minor concerns about your revised manuscript. Please revise your manuscript carefully again.

We would appreciate receiving your revised manuscript by Feb 14 2020 11:59PM. To enhance the reproducibility of your results, we recommend that if applicable you deposit your laboratory protocols in protocols.io, where a protocol can be assigned its own identifier (DOI) such that it can be cited independently in the future. For instructions see: http://journals.plos.org/plosone/s/submission-guidelines#loc-laboratory-protocols

We look forward to receiving your revised manuscript.

Kind regards,

Kenji Hashimoto, PhD

Academic Editor

PLOS ONE

Reviewers' comments:

Reviewer's Responses to Questions

**Comments to the Author**

1. If the authors have adequately addressed your comments raised in a previous round of review and you feel that this manuscript is now acceptable for publication, you may indicate that here to bypass the “Comments to the Author” section, enter your conflict of interest statement in the “Confidential to Editor” section, and submit your "Accept" recommendation.

Reviewer #1: (No Response)

Reviewer #2: All comments have been addressed

2. Is the manuscript technically sound, and do the data support the conclusions?

Reviewer #1: Partly

Reviewer #2: Yes

3. Has the statistical analysis been performed appropriately and rigorously? 

Reviewer #1: Yes

Reviewer #2: Yes

4. Have the authors made all data underlying the findings in their manuscript fully available?

Reviewer #1: No

Reviewer #2: Yes

5. Is the manuscript presented in an intelligible fashion and written in standard English?

Reviewer #1: Yes

Reviewer #2: Yes

6. Review Comments to the Author

Reviewer #1: I have gone through the revision of he manuscript entitled Association of overtime work hours with various stress responses in 59,021 Japanese workers: retrospective cross-sectional study. I think the authors were very responsive to the reviewer comments. In my view, the manuscript in its current form has the potential to make a meaningful contribution to the literature on overtime. Below I outline a couple of minor issues that ideally should be addressed to make the manuscript more reader-friendly.

1. On page 8 and 9 you state that you combined the categories 70-80 hours and 81+ hours to one category. To me it seems like this decision is not reflected in all the analyses presented. I suppose this statement occurs erroneously. Please align the statements in the methods section with the focal analyses.

2. I suggest you introduce the focal stress responses in the theory section. At a minimum, you should give readers (unfamiliar with the specific literature on overtime) an idea which facets of experienced stress you study why combining these variables makes sense. In response to one of my comments to added a discussion on work engagement, vigor etc. I suggest stating upfront which facets of individual stress responses your study is aimed at.

3. On a related note, in response to my comment in the last review, you added a section to argue for the differences between vigor as a facet of work engagement and vigor as a facet affect in terms of the POMS. According to this section both conceptualizations are different because they have been conceptualized by different authors or in different domains. In my view, such a statement is unfortunate for two reasons: First, it does not clarify what the actual differences are. Second, it is not a strong argument for distinguishing the two conceptualizations of vigor. I think, Shirom has worked out the conceptual overlap and distinctiveness between energy-related constructs in several chapters and papers. Just one idea: The vigor facet of work engagement is explicitly work-related. The vigor facet of the POMS is more generic. I don’t think it is necessary to call for future research on the work engagement vigor facet. However, I urge you to be precise regarding the definition and the conceptualization of the stress responses. Please state clearly upfront (in the theory section) what defines fatigue, vigor, depressive mood etc. and why specifically these stress responses do make sense as a bundle. I would explain at the beginning what vigor and fatigue have in common and to what extent they are different. Please note that this is a conceptual rather than an empirical question. You might find the literature on human energy as a unifying framework helpful (Quinn, Spreitzer, & Lam, 2012).

4. Please be more specific in the methods sections which stress response has been derived from the POMS and which has been derived from the other scales.

5. In my last review, I encouraged thinking of ways how your study might provide a theoretical contribution (besides the more descriptive contribution). My sense is that you have basically ignored my advice and reiterate over and over again your initial idea of doing an analysis of overtime and stress responses including more women. I have been taught not trying to make others write the paper I would like to read or I would like to write myself. Therefore, my advice for future papers would be: Take constructive advice seriously and consider thinking outside the box. Your sample size is much larger than in any meta-analysis on any subject I know. However, instead of scooping the potential inherent in the data to inform theory building to integrate research on overtime etc., you do descriptive piece on whether different amounts of relate differently to a set of stress-responses. Given that this is exactly what PLOSone embraces, there is no need to respond to this comment of mine.

6. I appreciate the clarification regarding the analyses. I think the analyses treating overtime as a continuous variable is most relevant. Consider making this analysis more focal. My impression is that – in the current version of the manuscript – you emphasize the regression models applying the overtime categories as dummy variables. For instance, in Table 2 you examine over and over again, whether 61 hours of overtime is different from less than 20 hours and whether 41 hours are different from less than 20 hours. I think the linear analysis and the dummy analyses test different hypotheses: Linear change vs. change after crossing a threshold of 21 hours. In my view, there are two ways to deal with this issue: First, make the threshold analyses less focal to the manuscript. Second, provide a rationale for conducting these competing views. Again, this would be very interesting and relevant from a theoretical perspective – although theoretical considerations don’t seem to be your primary aim with this manuscript. At a minimum, I would explain the rationale of the two competing views and the two sets of analyses more clearly in the paper – probably mentioning this aspect in the theory section already.

7. There is something like “lineally” in the manuscript. I am not sure whether this expression actually exists in English. Probably, “linearly” would be a better fit if you mean to refer to linear trajectories.

Reviewer #2: Manuscript Number: PONE-D-19-24855R1

Manuscript Title: Association of overtime work hours with various stress responses in 59,021 Japanese workers: retrospective cross-sectional study

All my comments raised in the first round of review were adequately addressed and I thank the authors for having carefully revised the manuscript. In my judgement this manuscript is now acceptable for publication in PLOS ONE.

7. PLOS authors have the option to publish the peer review history of their article (what does this mean?). If published, this will include your full peer review and any attached files.

Reviewer #1: Yes: Oliver Weigelt

Reviewer #2: No

---

## [Author Response · Author response to Decision Letter 1]

30 Jan 2020

Reviewer #1

I have gone through the revision of he manuscript entitled Association of overtime work hours with various stress responses in 59,021 Japanese workers: retrospective cross-sectional study. I think the authors were very responsive to the reviewer comments. In my view, the manuscript in its current form has the potential to make a meaningful contribution to the literature on overtime. Below I outline a couple of minor issues that ideally should be addressed to make the manuscript more reader-friendly.

Comment #1

On page 8 and 9 you state that you combined the categories 70-80 hours and 81+ hours to one category. To me it seems like this decision is not reflected in all the analyses presented. I suppose this statement occurs erroneously. Please align the statements in the methods section with the focal analyses.

Our response for this comment

We apologize this confusing notation. We used eight categories for overtime, i.e.“20 hours or less”, “21-30 hours”, “31-40 hours”, “41-50 hours”, “51-60 hours”, “61-70 hours”, “71-80 hours” and “81 hours or more”. The category of “81+” was generated after consolidating all categories from “81-90 hours“ to “140+ hours”. We revised the manuscript to clarify this point. 

(Method section, page 8-9, line 122-127) 

This study assessed the length of participants’ monthly length of overtime working hours. Self-reported monthly overtime data was collected in increments of every 10 hours from “20 hours or less” to “141 hours or more”. Due to the smaller number of participants who engaged in 81 hours or more overtime we consolidated them into one category, and thus the present study set the categorization of overtime as the following 8 categories, “20 hours or less”, “21-30 hours”, “31-40 hours”, “41-50 hours”, “51-60 hours”, “61-70 hours”, “71-80 hours” and “81 hours or more”.

Comment #2&3

I suggest you introduce the focal stress responses in the theory section. At a minimum, you should give readers (unfamiliar with the specific literature on overtime) an idea which facets of experienced stress you study why combining these variables makes sense. In response to one of my comments to added a discussion on work engagement, vigor etc. I suggest stating upfront which facets of individual stress responses your study is aimed at.

On a related note, in response to my comment in the last review, you added a section to argue for the differences between vigor as a facet of work engagement and vigor as a facet affect in terms of the POMS. According to this section both conceptualizations are different because they have been conceptualized by different authors or in different domains. In my view, such a statement is unfortunate for two reasons: First, it does not clarify what the actual differences are. Second, it is not a strong argument for distinguishing the two conceptualizations of vigor. I think, Shirom has worked out the conceptual overlap and distinctiveness between energy-related constructs in several chapters and papers. Just one idea: The vigor facet of work engagement is explicitly work-related. The vigor facet of the POMS is more generic. I don’t think it is necessary to call for future research on the work engagement vigor facet. However, I urge you to be precise regarding the definition and the conceptualization of the stress responses. Please state clearly upfront (in the theory section) what defines fatigue, vigor, depressive mood etc. and why specifically these stress responses do make sense as a bundle. I would explain at the beginning what vigor and fatigue have in common and to what extent they are different. Please note that this is a conceptual rather than an empirical question. You might find the literature on human energy as a unifying framework helpful (Quinn, Spreitzer, & Lam, 2012). 

Our response for these comments

(Since these comments related with each other, we would like to respond together.)

First of all, we appreciate this thoughtful comment. As you pointed, theoretical consideration would be needed to interpret the findings in this paper. With regards to difference between vigor and fatigue, we thought the correlation between these two factors is informative for readers, so we added that information in results section. Furthermore we added some explanation in discussion section. These are from our thought that we would like this paper to put more focus on epidemiological contribution. 

(Discussion part, Page 30-31, Line 341-352)

Vigor among workers have attracted research attention in positive occupational health psychology for instance in studies of work engagement (50,51). Work engagement has been defined as “a positive, fulfilling, work-related state of mind that is characterized by vigor, dedication and absorption”(52). Vigor in work-engagement assessments has included concepts of “motivation” or “psychological resilience”, whereas the items for assessment of vigor in BJSQ have been based on POMS. According to Shirom, the conceptualization of vigor in both work-engagement and BJSQ is not exactly the same (53). In our study “lack of vigor” was measured from the POMS. This negative concept is related on a deeper theoretical level to fatigue. In fact, moderate correlations between “lack of vigor” and “fatigue” were observed (men: r=-0.328, women: r=-0.358). However, the distinction between these interrelated concepts remains unclear theoretically. Exploring the association between overtime working hours and vigor in work engagement is an important research question in future studies

Comment #4

Please be more specific in the methods sections which stress response has been derived from the POMS and which has been derived from the other scales.

Our response for this comment

We appreciate this comment. We added detailed explanation how stress-response scales in the BJSQ were chosen into six scales.

(Method section, page 9, line 130-140)

Stress responses were assessed using the Brief Job Stress Questionnaire (BJSQ) following the recommended protocol of the Stress Check Program (25–27). The BJSQ was originally designed to measure both psychological and somatic, both positive and negative stress responses among workers in any workplace with minimum number of items (28). Six stress-response scales can be measured by 29 questionnaire items in the BJSQ, and each items were developed by referring some already standardized/authorized questionnaires. In detail, “vigor”, “fatigue”, and “irritability” , consisting of 3 items each, were from the Profile Of Mood States (POMS). “Depression”, consisting of 6 items, was from the Center for epidemiologic Studies for Depression Scale (CES-D). “Anxiety”, consisting of 3 items, was from the State-Trail Anxiety Inventory (STAI). “Somatic stress responses”, consisting of 11 items, was from Screener for the Somatoform Disorders and the Subjective Wellbeing Inventory (SUBI)(28).

Comment #5

I appreciate the clarification regarding the analyses. I think the analyses treating overtime as a continuous variable is most relevant. Consider making this analysis more focal. My impression is that – in the current version of the manuscript – you emphasize the regression models applying the overtime categories as dummy variables. For instance, in Table 2 you examine over and over again, whether 61 hours of overtime is different from less than 20 hours and whether 41 hours are different from less than 20 hours. I think the linear analysis and the dummy analyses test different hypotheses: Linear change vs. change after crossing a threshold of 21 hours. In my view, there are two ways to deal with this issue: First, make the threshold analyses less focal to the manuscript. Second, provide a rationale for conducting these competing views. Again, this would be very interesting and relevant from a theoretical perspective – although theoretical considerations don’t seem to be your primary aim with this manuscript. At a minimum, I would explain the rationale of the two competing views and the two sets of analyses more clearly in the paper – probably mentioning this aspect in the theory section already.

Our response for this comment

We appreciate this valuable comment. We agree with adding explanation why we use two statistical models simultaneously. Thus, we added explanations in the manuscript. 

(Method section page 11-12, line 160-167)

Multiple linear regression analysis was used to examine the associations of stress responses with overtime. In each model, we used a standardized score of each stress response (such as “lack of vigor” or “irritability”, etc.) as a dependent variable, “length of overtime work hours” as an independent variable, other individual characteristics (such as “age” and “job position”, etc.) as covariates. First, we performed linear trend tests by treating “length of overtime work hours” as a continuous variable to check the linear association between each stress response and overtime. Then, to seek possible thresholds of overtime, “length of overtime work hours” were treated as a dummy variable by setting “less than 20hours” group as a reference category.

Comment #6

There is something like “lineally” in the manuscript. I am not sure whether this expression actually exists in English. Probably, “linearly” would be a better fit if you mean to refer to linear trajectories.

Our response for this comment

We apologize this mistake, thus we revised the part.

(Abstract section, page 3, line 45-46)

Length of overtime was linearly associated with various stress responses, except for “lack of vigor”.

---

## [Decision Letter · Decision Letter 2]

10 Feb 2020

Association of overtime work hours with various stress responses in 59,021 Japanese workers: retrospective cross-sectional study

PONE-D-19-24855R2

Dear Dr. Odagiri,

We are pleased to inform you that your manuscript has been judged scientifically suitable for publication and will be formally accepted for publication once it complies with all outstanding technical requirements.

With kind regards,

Kenji Hashimoto, PhD

Section Editor

PLOS ONE

Additional Editor Comments (optional):

Reviewers' comments:

Reviewer's Responses to Questions

**Comments to the Author**

1. If the authors have adequately addressed your comments raised in a previous round of review and you feel that this manuscript is now acceptable for publication, you may indicate that here to bypass the “Comments to the Author” section, enter your conflict of interest statement in the “Confidential to Editor” section, and submit your "Accept" recommendation.

Reviewer #1: All comments have been addressed

2. Is the manuscript technically sound, and do the data support the conclusions?

Reviewer #1: Yes

3. Has the statistical analysis been performed appropriately and rigorously? 

Reviewer #1: Yes

4. Have the authors made all data underlying the findings in their manuscript fully available?

Reviewer #1: No

5. Is the manuscript presented in an intelligible fashion and written in standard English?

Reviewer #1: Yes

6. Review Comments to the Author

Reviewer #1: I think you have addressed the most important issues from my last review. Of note, you have done a good job clarifying the aspects that needed clarification. I ask you for one more thing:

Please include a correlation table including all relevant variables (dependent, independent, covariates) for the focal sample (no separate correlations per sex needed here). Inclusion of the correlation table is important for researchers doing meta-ananlyses on one or several of the variables studied. I am sorry, I ask you for this so late in the process. Your mention of the correlations between vigor and fatigue, made me aware that reporting the correlations is mandatory for most survey studies.Please also include a measure of reliability (Cronbachs Alpha or McDonalds Omega). These pieces of information are important for meta-analyses, too.

7. PLOS authors have the option to publish the peer review history of their article (what does this mean?). If published, this will include your full peer review and any attached files.

Reviewer #1: Yes: Oliver Weigelt

---

## [Editor Report · Acceptance letter]

14 Feb 2020

PONE-D-19-24855R2 

Association of overtime work hours with various stress responses in 59,021 Japanese workers: retrospective cross-sectional study 

Dear Dr. Odagiri:

I am pleased to inform you that your manuscript has been deemed suitable for publication in PLOS ONE. Congratulations! Your manuscript is now with our production department. 

With kind regards,

on behalf of

Prof. Kenji Hashimoto 

Section Editor

PLOS ONE